# Auditory input enhances somatosensory encoding and tactile goal-directed behavior

L. Godenzini[1], D. Alwis[1], R. Guzulaitis[1], S. Honnuraiah[2], G. J. Stuart [2] & L. M. Palmer [1✉]

The capacity of the brain to encode multiple types of sensory input is key to survival. Yet, how neurons integrate information from multiple sensory pathways and to what extent this influences behavior is largely unknown. Using two-photon $Ca^{2+}$ imaging, optogenetics and electrophysiology in vivo and in vitro, we report the influence of auditory input on sensory encoding in the somatosensory cortex and show its impact on goal-directed behavior. Monosynaptic input from the auditory cortex enhanced dendritic and somatic encoding of tactile stimulation in layer 2/3 (L2/3), but not layer 5 (L5), pyramidal neurons in forepaw somatosensory cortex (S1). During a tactile-based goal-directed task, auditory input increased dendritic activity and reduced reaction time, which was abolished by photo-inhibition of auditory cortex projections to forepaw S1. Taken together, these results indicate that dendrites of L2/3 pyramidal neurons encode multisensory information, leading to enhanced neuronal output and reduced response latency during goal-directed behavior.

[1] Florey Institute of Neuroscience and Mental Health, University of Melbourne, Melbourne, VIC, Australia. [2] Eccles Institute of Neuroscience and ANU node of the Australian Research Council Centre of Excellence for Integrative Brain Function, John Curtin School of Medical Research, Australian National University, Canberra, ACT, Australia. ✉email: lucy.palmer@florey.edu.au

One of the fundamental tasks of the brain is to make sense of the world. This requires the concurrent integration of different sensory information streams, which are constantly changing as an animal moves through its environment. During sensory-guided behaviors, the integration of multiple sensory pathways (multisensory integration) typically results in shorter reaction times, lower sensory-detection thresholds, and enhanced object recognition[1–7]. Such changes in behavior are believed to be driven, in part, by the modulation of sensory processing within the cortex[8].

Consistent with the complex connectivity within the brain[9], there is an increasing body of literature demonstrating the convergence of multisensory projections in primary sensory cortex[10–12]. For example, in the visual cortex, multisensory input results in altered encoding and sensory processing[13–15]. Despite the known influence of multisensory input on overall neural activity, information about the cellular mechanisms leading to changes in cortical processing and sensory-based behavior is limited. In the cortex, the integration of multisensory information requires the combination of top-down input from different sensory brain regions with the bottom-up input from the primary sense[16]. This integration can occur at the level of single cortical neurons, often resulting in dendritic electrogenesis[17].

Dendritic integration ultimately influences neuronal firing[16, 18–25], which is crucial for sensory encoding and perception of a single sense[26, 27]. However, our external world is sensory-rich and therefore individual sensory modalities are usually not sensed in isolation. Cortical layer 2/3 (L2/3) pyramidal neurons are perfectly positioned to act as a hub for encoding of multisensory information. They directly influence cortical output neurons[28–30], and also send and receive long-range projections from other cortical and subcortical areas[31, 32]. However, how individual L2/3 pyramidal neurons integrate signals from multiple sensory pathways and the impact this has on somatic output is largely unknown. Here, we used a combination of $Ca^{2+}$ imaging, patch-clamp electrophysiology, and optogenetics during goal-directed behavior to investigate the influence of the activation of one sense, in this case the auditory system, on sensory processing in the primary somatosensory cortex. We find that the integration of auditory and somatosensory input in tuft dendrites of L2/3, but not L5, pyramidal neurons in forepaw S1 leads to enhanced sensory encoding and faster reaction times during tactile goal-directed behavior.

## Results

### Auditory stimuli enhance sensory encoding in tuft dendrites of L2/3 pyramidal neurons in S1.
To assess the influence of auditory input on somatosensory processing in S1, we first characterized whether the auditory cortex sends direct axonal projections to forepaw S1. Using dual fluorescence imaging to label both auditory cortex projections and the somatosensory cortex, we visualized a high density of axons projecting from the auditory cortex within forepaw S1 (Supplementary Fig. 1). These auditory axons were predominantly restricted to the upper cortical layers of forepaw S1 (Fig. 1a). Since the tuft dendrites of pyramidal neurons stratify in the upper layers of the cortex, we investigated the influence of auditory input on somatosensory processing in the tuft dendrites of pyramidal neurons in the forepaw S1 of awake mice (Fig. 1a). We first targeted L2/3 pyramidal neurons as they send and receive projections from other cortical and subcortical areas[32], making them ideally positioned for encoding multisensory information. L2/3 pyramidal neurons were sparsely transfected with the $Ca^{2+}$ indicator, GCaMP6f, and following expression, $Ca^{2+}$ activity was recorded from tuft dendrites (higher-order branches) that were located between 50 and 100 µm below the pia using two-photon microscopy. In naive mice, tactile stimulation of the contralateral forepaw (200 Hz, 500 ms) evoked large $Ca^{2+}$ transients (>3 s.d. of the baseline fluorescence) in L2/3 tuft dendrites (Fig. 1b; 2.39 ± 0.14 ΔF/F; n = 110 dendrites, 11 mice). To test the influence of auditory input, mice were presented with a broadband auditory stimulus (Aud, 2–50 kHz, 75 dB, 500 ms), which evoked activity in the auditory cortex (Supplementary Fig. 2). Surprisingly, this broadband auditory stimulus also evoked $Ca^{2+}$ transients in tuft dendrites of L2/3 pyramidal neurons in forepaw S1 (Fig. 1b; 2.86 ± 0.15 ΔF/F; n = 110 dendrites, 11 mice). In fact, in L2/3 pyramidal neuron tuft dendrites that were also active during tactile stimulus, the auditory-evoked $Ca^{2+}$ transients were significantly larger than those evoked by tactile stimulation alone (Fig. 1c; p = 0.0014). Furthermore, when auditory and tactile stimulus were presented simultaneously (AudTac), there was a further increase in the amplitude of evoked $Ca^{2+}$ responses (Fig. 1b, c; 3.29 ± 0.16 ΔF/F; n = 110 dendrites, 11 mice; p < 0.0001). Likewise, there was also a significant increase in the rate of evoked $Ca^{2+}$ responses during paired tactile and auditory stimulus (Fig. 1d; Tac, 0.09 ± 0.008; AudTac, 0.12 ± 0.01; n = 110 dendrites, 11 mice; p < 0.001). This influence of auditory input on the amplitude of evoked $Ca^{2+}$ transients in tuft dendrites did not depend on the tactile-stimulus duration nor auditory-stimulus intensity (Supplementary Fig. 3). In contrast, the rate of tactile-evoked $Ca^{2+}$ transients was influenced by the intensity of auditory broadband stimulus, with lower auditory intensities having a greater influence on dendritic encoding (Supplementary Fig. 3). Additionally, auditory-evoked $Ca^{2+}$ transients were not due to large body movements associated with a startle response as the presentation of the auditory stimulus was not correlated with movement detected by EMG recordings from the neck muscles (Supplementary Fig. 4), however, it must be noted that it is possible that undetected subtle body movements may still occur in response to auditory stimuli. Furthermore, pupil dilation was similar during auditory and tactile stimuli (Aud, 0.056 ± 0.02 mm; Tac, 0.059 ± 0.02 mm; n = 6 mice, p = 0.56), suggesting the stimuli are of comparable saliency and had a similar influence on the arousal state (Supplementary Fig. 5). These data indicate that when paired together, the auditory stimulus increased the dendritic response to tactile stimuli, highlighting a potential role of auditory input to forepaw S1 in the dendritic integration of multisensory information.

### Auditory stimuli enhance tactile-evoked action potentials in L2/3 pyramidal neurons in S1.
Does the enhanced multisensory integration in tuft dendrites during auditory input influence somatic output and the transfer of somatosensory information in L2/3 pyramidal neurons? To investigate this, whole-cell patch-clamp recordings were made from the soma of L2/3 pyramidal neurons in forepaw S1 of awake mice (Fig. 2a). In response to tactile stimulation of the forepaw, L2/3 pyramidal neurons reliably evoked action potentials above baseline (Fig. 2b; Baseline, 1.05 ± 0.23 Hz vs Tactile, 2.29 ± 0.59 Hz; n = 17 neurons, 8 mice; p = 0.027). Likewise, when tactile stimulation was paired with an auditory stimulus, the evoked firing rate was also significantly increased above baseline (Fig. 2c; Baseline, 1.11 ± 0.22 Hz vs AudTac, 2.61 ± 0.63 Hz; n = 17 neurons, 8 mice; p = 0.008). This is in contrast to the evoked firing rate in response to an auditory stimulus on its own, which did not differ significantly from baseline (Fig. 2d; Baseline, 1.16 ± 0.27 Hz vs Aud, 1.20 ± 0.31 Hz; n = 17 neurons, 8 mice; p = 0.75). These data suggest that auditory input alone does not result in action potential generation,

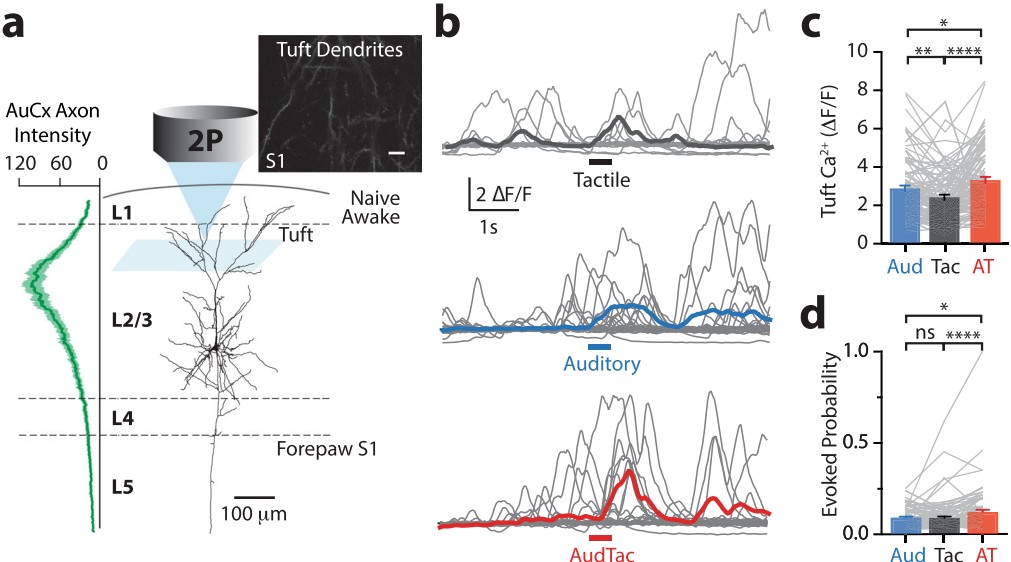

**Fig. 1 Auditory stimulus enhances sensory encoding in tuft dendrites of layer 2/3 pyramidal neurons in the primary somatosensory cortex. a** (Left) The maximum relative optical density of axonal projections from the auditory cortex (AuCx) within the forepaw primary somatosensory cortex (S1) ($n = 3$ mice). (Right) Schematic of experimental design. In vivo two-photon $Ca^{2+}$ imaging was performed in the tuft dendrites of L2/3 pyramidal neurons previously injected with the genetic $Ca^{2+}$ indicator, GCaMP6f. Inset, two-photon image of tuft dendrites recorded 90 μm below pia. Inset scale, 10 μm. **b** Typical evoked $Ca^{2+}$ responses recorded in an example tuft dendrite during tactile (black), auditory (blue) and paired tactile and auditory (red) stimuli. Colored thick line, average response. **c** The amplitude of sensory-evoked $Ca^{2+}$ transients during tactile (black), auditory (blue) and paired tactile and auditory (red) stimuli (One-way ANOVA Friedman test, $p < 0.0001$; Dunn's multiple comparisons test, $n = 110$ dendrites, 11 mice; $p < 0.0001$, $p = 0.001$, $p = 0.037$). **d** The rate of sensory-evoked $Ca^{2+}$ transients during tactile (black), auditory (blue) and paired tactile and auditory (red) stimuli (One-way ANOVA Friedman test, $p < 0.0001$; Dunn's multiple comparisons test, $n = 110$ dendrites, 11 mice; $p < 0.0001$, $p = 0.191$, $p = 0.023$). Error bars represent S. E.M. *$p < 0.05$, **$p < 0.01$, ****$p < 0.0001$.

and that co-activation with other pathways are required to influence somatic output[24, 33]. Indeed, when the tactile stimulus was paired with an auditory stimulus there was a significant increase in the evoked firing rate compared to the tactile stimulus alone (Fig. 2e; $n = 16$ neurons, 8 mice; $p = 0.013$). Similar results were obtained during two-photon $Ca^{2+}$ imaging from the somas of L2/3 pyramidal neurons, where there was a significant increase in the probability of $Ca^{2+}$ transients evoked during paired tactile and auditory stimuli compared to tactile stimulus alone (Supplementary Fig. 6; $p = 0.0001$, $n = 127$ somas, 4 mice). This enhanced somatic output during paired auditory and tactile input was not reflected in an increase in the somatic membrane potential. Both tactile alone and paired tactile and auditory stimuli evoked robust voltage responses at the soma, which had similar amplitude (Fig. 2f; $7.1 \pm 1.0$ mV vs $7.2 \pm 0.8$ mV; $n = 17$ neurons, 8 mice; $p = 0.644$). Similar results were also obtained in L2/3 pyramidal neurons in the anaesthetized state, where paired tactile and auditory simulation lead to a similar subthreshold voltage response to those evoked by tactile stimulation alone ($18.4 \pm 1.6$ mV vs $18.4 \pm 1.7$ mV; $p = 0.525$), but significantly increased the number of evoked action potentials (Supplementary Fig. 7; $p = 0.049$; $n = 15$ neurons, 12 mice). Therefore, as reported previously[34], the increase in firing during paired auditory and tactile stimulus is not simply due to increased summation of synaptic input at the soma. Combining the dendritic (Fig. 1) and somatic (Fig. 2) results, the negligible influence at the soma of auditory stimuli on its own suggests that auditory input specifically enhances local dendritic activity. Indeed, two-photon $Ca^{2+}$ imaging from the tuft dendrites and somatas of the same population of L2/3 pyramidal neurons illustrate that auditory-evoked $Ca^{2+}$ responses in tuft dendrites were significantly greater than somatic responses (Supplementary Fig. 6; tuft, $1.04 \pm 0.09$ vs soma, $0.55 \pm 0.2$; $p < 0.0001$; $n = 116$ tuft dendrites, 38 soma).

Taken together, these results suggest that activation of one primary sensory area can alter the somatic output in another primary sensory area during sensory encoding.

**Auditory cortex directly projects to L2/3, not L5, pyramidal neurons in S1.** Is the influence of auditory input on somatosensory encoding due to direct monosynaptic connectivity between forepaw S1 and auditory cortex? To examine this, we turned to the brain slice preparation where synaptic inputs and cellular excitability can be precisely investigated. The photo-active opsin ChR2 (AAV1.hSyn.ChR2(H134R)-eYFP.WPRE.hGH) was injected into the auditory cortex (200–500 μm below pia) and following expression, somatic whole-cell recordings were made from L2/3 pyramidal neurons in forepaw S1 in vitro (Fig. 3a). Auditory cortex axons containing ChR2 were activated by brief LED pulses (470 nm; 2 ms) and slices were bathed in TTX plus 4-AP to determine if evoked responses were monosynaptic (Supplementary Fig. 8). Photoactivation of axonal projections from auditory cortex generated robust EPSPs in 76% (22/29 neurons) of L2/3 pyramidal neurons in forepaw S1, which increased in amplitude as the intensity of the LED stimulus was increased (Fig. 3b; 0.2–1.0 mW; $n = 16$ neurons). These responses were abolished by inclusion of DNQX and AP5 in the bath, illustrating they were mediated by AMPA/NMDA receptors (Fig. 3c) and remained in the presence of TTX plus 4-AP, indicating that they are monosynaptic (Supplementary Fig. 8). These data indicate that the auditory cortex provides direct synaptic input onto L2/3 pyramidal neurons in forepaw S1. To investigate the influence this monosynaptic auditory input has on firing properties of L2/3 pyramidal neurons, brief somatic current pulses (100 ms) were paired with LED activation (Fig. 3d). Photoactivation of auditory cortex axons significantly decreased rheobase (Fig. 3e; control, $291 \pm 30$ pA; LED, $208 \pm 55$ pA; $n = 6$ neurons, $p = 0.03$),

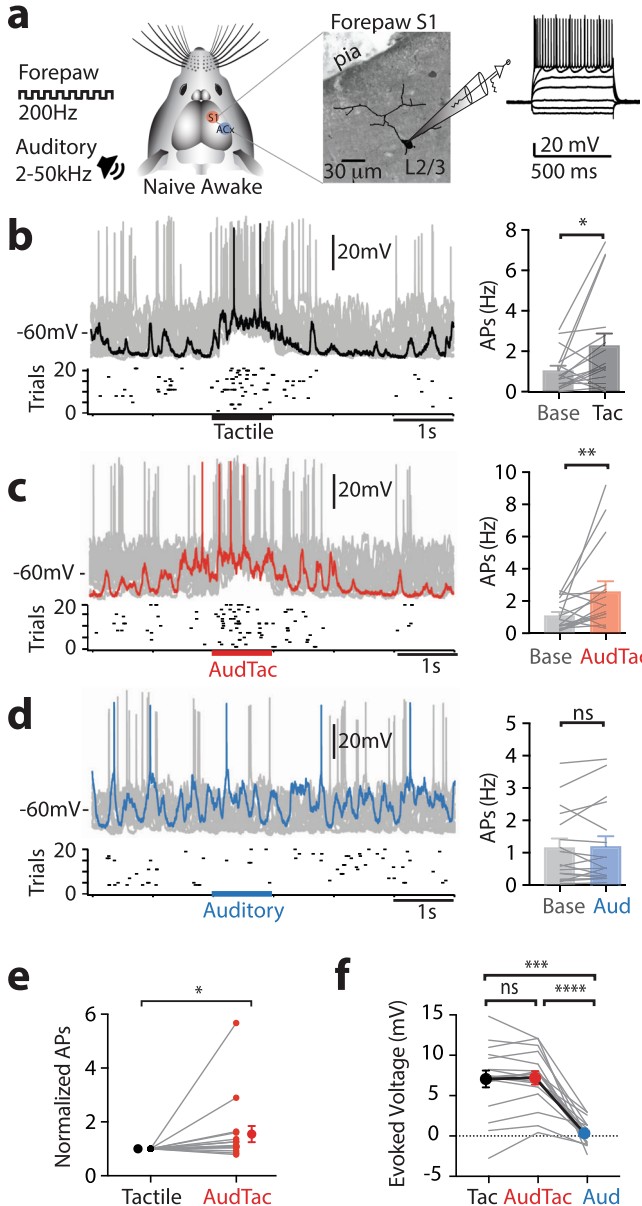

**Fig. 2 Auditory input enhances tactile-evoked somatic action potentials in layer 2/3 pyramidal neurons in vivo. a** Left, Schematic of experimental design. Mice were previously habituated to head fixation before whole-cell patch-clamp recordings were performed from L2/3 pyramidal neurons in forepaw S1 in the awake state. Right, L2/3 pyramidal neuron filled with fluorescence biocytin for reconstruction and example voltage response to injected current steps (50 pA). (Left) Overlay of individual trials and raster of action potentials from a typical neuron and (right) average evoked action potentials per trial in all recorded neurons in response to (**b**) tactile stimulus (Wilcoxon matched-pairs signed rank test; $p = 0.027$; $p_{shuffled} = 0.782$; 27 trials$_{av}$), (**c**) paired auditory and tactile stimulus (two-tailed Wilcoxon matched-pairs signed rank test; $p = 0.008$; $p_{shuffled} = 0.927$; 28 trials$_{av}$) and (**d**) auditory stimulus (two-tailed Wilcoxon matched-pairs signed rank test, $p = 0.75$; $p_{shuffled} = 0.41$; 21 trials$_{av}$). $n = 17$ neurons, 8 mice. Base = baseline. Colored trace, single example voltage traces for tactile (black), paired auditory and tactile (red), auditory (blue) stimulus. **e** Average evoked action potentials during paired auditory and tactile stimulus (red) normalized to tactile stimulus (black) ($p = 0.013$). Wilcoxon matched-pairs signed rank test. $n = 17$ neurons, 8 mice. **f** Mean somatic voltage response during tactile (black), paired auditory and tactile (red), and auditory (blue) stimulus ($n = 17$ neurons, 8 mice; One-way ANOVA Friedman test, $p < 0.0001$; Dunn's multiple comparisons test. $p = 0.0002$, $p > 0.999$, $p < 0.0001$). Error bars represent S.E.M. *$p < 0.05$, **$p < 0.01$, ***$p < 0.001$, ****$p < 0.0001$.

neurons were absent because the apical dendrites had been severed during the slicing procedure, the dendritic tree of recorded L5 pyramidal neurons were morphologically reconstructed and shown to be intact (Supplementary Fig. 9). Together, these data indicate that axonal projections from auditory cortex to S1 make monosynaptic connections with L2/3, but not L5, pyramidal neurons in forepaw S1.

**Auditory input does not influence dendritic integration and somatic output in L5 pyramidal neurons.** While L5 pyramidal neurons do not receive direct monosynaptic input from the auditory cortex, we nevertheless tested whether auditory input can indirectly influence the encoding of somatosensory information in L5 pyramidal neurons during a forepaw stimulus in vivo. Using two-photon Ca$^{2+}$ imaging, responses to sensory input were recorded in awake mice sparsely transfected with the Ca$^{2+}$ indicator GCaMP6f in L5 of forepaw S1 (Fig. 4a). Large Ca$^{2+}$ transients were evoked in tuft dendrites of L5 pyramidal neurons in response to tactile stimuli (Fig. 4a, b; $2.56 \pm 0.11$ $\Delta$F/F; $n = 112$ dendrites, 4 mice). Unlike L2/3 neurons, the amplitude of these tactile-evoked responses was not significantly altered when paired with auditory input (Fig. 4c; AudTac; $2.51 \pm 0.11$ $\Delta$F/F; $n = 106$ dendrites, 4 mice; $p = 0.794$). Ca$^{2+}$ transients were also evoked in the main apical dendrites of L5 pyramidal neurons in response to tactile input (Fig. 4d, e; $3.25 \pm 0.21$ $\Delta$F/F; $n = 44$ dendrites, 4 mice). Similar to L5 tuft dendrites, the amplitude of tactile-evoked responses in L5 apical dendrites were not significantly different when paired with auditory input (Fig. 4e, f; AudTac; $3.10 \pm 0.18$ $\Delta$F/F; $n = 44$ dendrites, 4 mice; $p = 0.755$). These results illustrate that auditory input does not influence sensory integration in L5 pyramidal neuron dendrites. Compared to L2/3 pyramidal neurons, the sensory-evoked Ca$^{2+}$ responses in the dendrites of L5 pyramidal neurons were significantly different in amplitude (Fig. 4g; $p < 0.0001$) and response probability ($p < 0.0001$; Fig. 4h). To test whether auditory input impacts the somatic output of L5 pyramidal neurons, we performed whole-cell recordings from the soma of L5 pyramidal neurons in forepaw S1 of awake mice (Fig. 4i). These experiments indicated that tactile stimulation of the forepaw alone reliably evoked a

however, had no effect on action potential threshold (Fig. 3f; Control, $-38.9 \pm 2.1$ mV; LED, $-40.7 \pm 1.6$ mV; $n = 6$ neurons, $p = 0.12$).

Since both L2/3 and L5 pyramidal neurons have their tuft dendrites stratifying in the upper layers of the cortex where auditory projections are predominately located, we also tested whether auditory cortex makes monosynaptic input onto L5 pyramidal neurons in forepaw S1. Whole-cell patch-clamp recordings were made from L5 pyramidal neurons in the same brain slices as L2/3 neurons to control for different axonal densities and/or expression in different slices/preparations (Fig. 3g). In contrast to L2/3 pyramidal neurons, photoactivation (470 nm; 2 ms) of auditory cortex projections did not generate significant somatic voltage responses in L5 pyramidal neurons in S1 (Fig. 3h, i; Small (<1 mV) responses where observed in 2/19 neurons). Consistent with this observation, photoactivation of auditory input during somatic current injection did not alter rheobase (Fig. 3j and k; Control, $529 \pm 51.7$ pA; LED, $517 \pm 47.9$ pA; $n = 9$ neurons, $p = 0.41$) or action potential threshold (Fig. 3l; Control, $-43.2 \pm 1.4$ mV; LED, $43.0 \pm 1.4$ mV; $n = 9$ neurons, $p = 0.56$). To rule out the possibility that voltage responses in L5

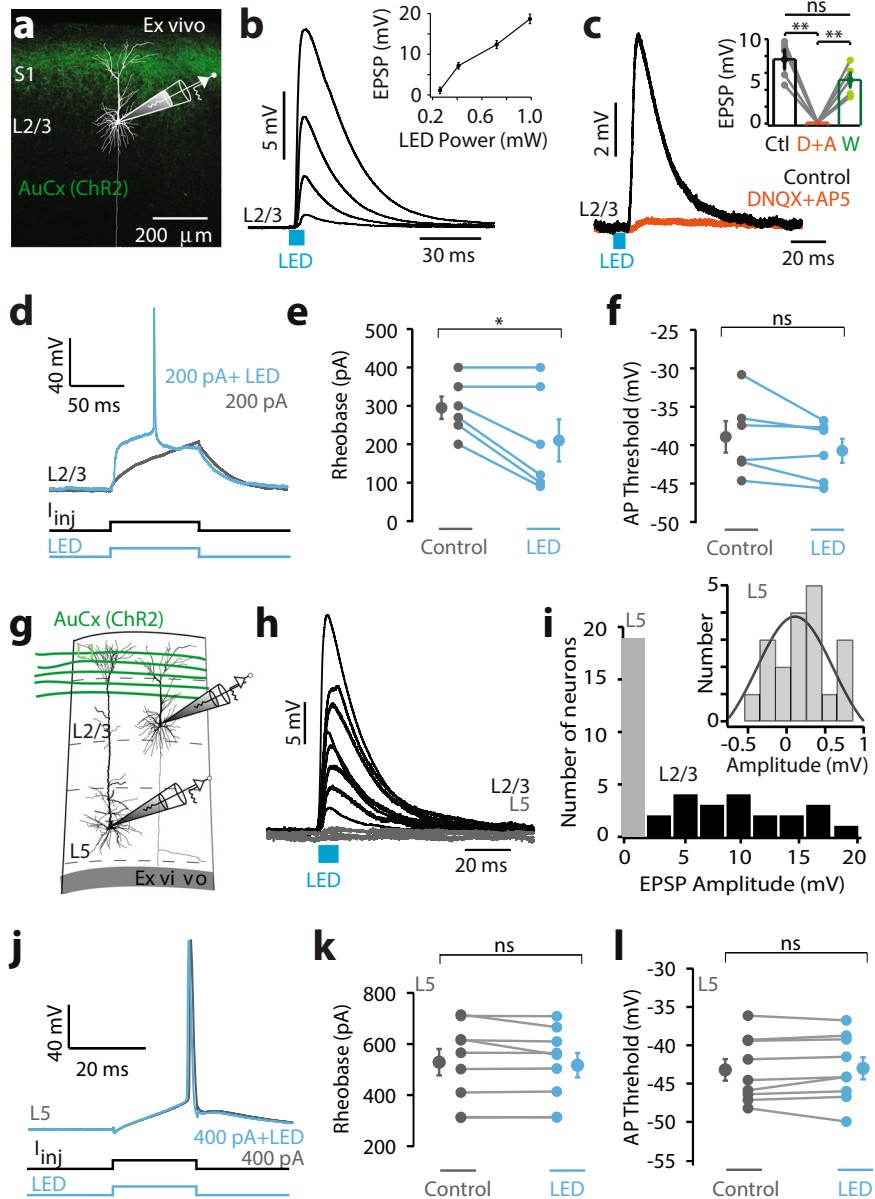

**Fig. 3 Auditory cortex directly projects to layer 2/3, and not layer 5, pyramidal neurons in the primary somatosensory cortex. a** Schematic of experimental paradigm. Photoactivatable opsin (ChR2) was injected into the auditory cortex (AuCx) and recordings were performed in the primary somatosensory cortex (S1). **b** Example of voltage responses in a L2/3 pyramidal neuron to photoactivation of auditory cortex axons with increasing LED power (470 nm, 2 ms, 0.16–5 mW). Inset, Average synaptic response to increasing LED ($n = 16$ neurons). **c** Example synaptic responses in a L2/3 pyramidal neuron to LED (470 nm; 2 ms) in control (black) and during bath application of DNQX + AP5 (orange). Inset, Average LED-evoked voltage responses in control (black), DNQX + AP5 (orange; $p = 0.0023$), and washout (green; $n = 5$ neurons; $p = 0.128$). **d** Example of voltage in a L2/3 pyramidal neuron to somatic current pulse alone (gray; 200 pA, 100 ms) and paired with LED (blue; 470 nm; 2 ms). **e** Rheobase during control current injection (gray) and paired with LED (blue; 470 nm; 2 ms) in L2/3 pyramidal neurons ($n = 6$ neurons; $p = 0.03$; $p_{shuffled} = 0.813$). **f** Action potential (AP) threshold during control somatic current injection (gray) and current injection paired with LED (blue; 470 nm; 2 ms) in L2/3 pyramidal neurons ($n = 6$ neurons; $p = 0.12$; $p_{shuffled} = 0.313$). **g** Patch-clamp recordings were performed in L2/3 and L5 pyramidal neurons in the same somatosensory brain slice. **h** Average light-evoked synaptic responses to photoactivation of auditory cortex axons (470 nm, 2 ms) in different L2/3 (black) and L5 (gray) pyramidal neurons. **i** Histogram of peak amplitudes of LED-evoked synaptic responses in L2/3 (black) and L5 (gray) pyramidal neurons. Inset, Histogram of L5 pyramidal neuron responses. **j** Same as (**d**) for a L5 pyramidal neuron. **k** Same as (**e**) for L5 pyramidal neurons ($n = 8$ neurons; $p = 0.41$; $p_{shuffled} = 0.297$). **l** Same as (**f**) for L5 pyramidal neurons ($n = 8$ neurons; $p = 0.56$; $p_{shuffled} = 0.57$). Two-tailed Wilcoxon matched-pairs signed rank test. Error bars represent S.E.M. *$p < 0.05$.

subthreshold voltage response (3.9 ± 0.62 mV) and action potential firing (Fig. 4j, k; 4.77 ± 1.20 Hz; $n = 16$ neurons, 5 mice). In contrast, auditory input did not significantly alter the tactile-evoked subthreshold voltage response (AudTac, 3.97 ± 0.61 mV; $n = 16$ neurons, 5 mice; $p = 0.43$) or action potential output (Fig. 4j, k; AudTac, 5.09 ± 1.15 Hz; $n = 16$ neurons, 5 mice; $p =$

0.81). Furthermore, auditory input had negligible impact on the voltage response at the soma (Fig. 4j; mean voltage, 0.63 ± 0.23 mV; $n = 16$ neurons, 5 mice;) and no significant impact on action potential output (Fig. 4j, k; baseline, 2.60 ± 0.65 Hz vs auditory, 2.85 ± 0.72 Hz; $n = 16$ neurons, 5 mice; $p = 0.83$). These results show that, unlike L2/3 pyramidal neurons, auditory input to

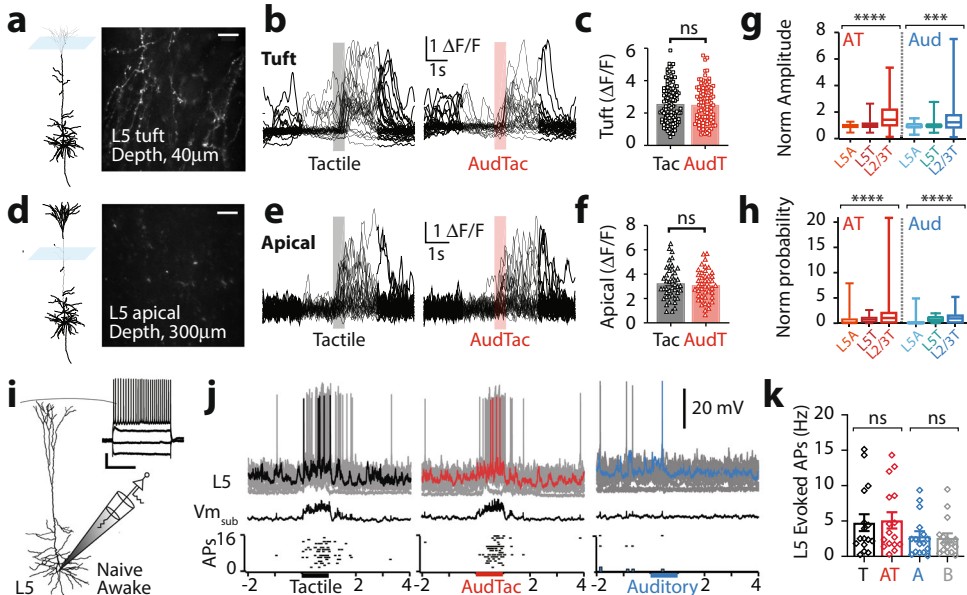

**Fig. 4 Auditory input does not influence dendritic integration, nor somatic output, in layer 5 pyramidal neurons.** Two-photon $Ca^{2+}$ imaging in the dendrites of L5 pyramidal neurons. **a** Example field of view. Imaging depth, 40 μm below pia. Scale bar, 10 μm. **b** $Ca^{2+}$ transients from a tuft dendrite shown in (**a**) during tactile stimulus alone (black) and paired with auditory stimulus (AudTac, red). **c** The amplitude of $Ca^{2+}$ transients in L5 tuft dendrites during tactile stimulus alone (black; $n = 112$ dendrites, 4 mice; 27 trials$_{av}$) and paired with auditory stimulus (red; $n = 106$ dendrites, 4 mice; 24 trials$_{av}$; $p = 0.794$; $p_{shuffled} = 0.960$; Mann–Whitney test). **d** Imaging depth, 300 μm below pia. **e** Same as (**b**) for L5 apical dendrites. **f** Same as (**c**) for L5 apical dendrites ($n = 44$ dendrites, 4 mice; 26 trials$_{av}$; $p = 0.755$; $p_{shuffled} = 0.931$; Mann–Whitney test). **g** Amplitude of all $Ca^{2+}$ transients in L5 apical (L5A; $n = 43/32$), L5 tuft (L5T; $n = 100/109$) and L2/3 tuft (L2/3 T; $n = 143/130$) dendrites during AudTac stimulus (red) and auditory stimulus (blue) normalized to the response to tactile stimulus. One-way ANOVA Friedman test, $p < 0.0001$, $p = 0.0005$. **h** Same as (**g**) for probability of evoked $Ca^{2+}$ transients. One-way ANOVA Friedman test, $p < 0.0001$, $p < 0.0001$. **i** Schematic of experimental paradigm. Patch-clamp recordings were performed from L5 pyramidal neurons in forepaw S1 from naive mice in the awake state. Inset, example voltage response to injected 50 pA steps of current. Scale, 20 mV, 400 ms. **j** (Top) Overlay of individual trials, (middle) average subthreshold voltage, and (bottom) raster of action potentials from a typical neuron in response to tactile stimulus (black), paired auditory and tactile stimulus (red) and auditory stimulus (blue). Colored trace, single example trace. **k** Evoked action potentials in response to tactile (black), paired auditory and tactile (red), auditory (blue) and baseline (gray) ($n = 16$ neurons, 5 mice; 17 trials$_{av}$) One-way ANOVA Friedman test. Error bars represent S.E.M. *$p < 0.05$.

forepaw S1 has minimal impact on dendritic activity and does not alter somatic action potential output of L5 pyramidal neurons.

**Auditory stimuli enhance dendritic tuft activity during tactile goal-directed behavior.** We next asked whether auditory input can impact somatosensory-based behavior. To investigate this, we trained mice in a tactile-based Go/NoGo goal-directed task (see methods). Here, mice rested their contralateral forepaw on a vibrating button which provided a non-noxious tactile vibration stimulus (200 Hz, 500 ms; Tactile-trial). If mice licked the sensor within 1 s of receiving the tactile stimulus, a sucrose water reward was delivered. Once trained in this task, mice were then randomly presented with an auditory stimulus (broadband noise, 2–50 kHz, 75 dB, 500 ms) either alone (Auditory-trial) or simultaneously with tactile stimulation (AudTac-trial). Mice were trained to ignore the auditory stimulus (Correct Rejection) and only received a water reward if they licked the sensor after the tactile stimulus was presented alone or when paired with the auditory stimulus (Hit) (Fig. 5a; see Methods). Mice rapidly learnt the task and were considered expert when the overall correct performance was greater than 80% (Supplementary Fig. 10). To test whether dendritic activity was altered by auditory input during sensory-based behavior, we performed $Ca^{2+}$ imaging in tuft dendrites of L2/3 pyramidal neurons in forepaw S1 while mice performed the goal-directed tactile task (Fig. 5b). Large $Ca^{2+}$ transients were evoked during rewarded trials (Hit) in ~40% of all active tuft dendrites (75/190 dendrites from 6 mice). During Tactile-trials, $Ca^{2+}$ transients were evoked in $15.7 \pm 2.1\%$ of correct trials (Fig. 5c). When paired with an auditory stimulus, the proportion

of tactile-trials with a $Ca^{2+}$ response was significantly increased to $21.1 \pm 2.6\%$ (Fig. 5d, e; $p < 0.0001$). The peak amplitude of $Ca^{2+}$ transients evoked in tuft dendrites with increased activity during AudTac-trials was also significantly increased by on average $71 \pm 31\%$ (Fig. 5f; $n = 58$ dendrites, 6 mice; $p = 0.0025$). Likewise, tactile-evoked $Ca^{2+}$ transients in L2/3 pyramidal neuron somata were also enhanced when paired with auditory stimuli in AudTac-trials during the tactile-based goal-directed task (Supplementary Fig. 11). This increase in neural activity during AudTac-trials compared to Tactile-trials was not due to differences in licking behavior, as there was no significant difference between the licking frequency during Tactile-trials and AudTac-trials ($3.1 \pm 0.4$ Hz vs $3.1 \pm 0.4$ Hz; $n = 16$ mice, $p = 0.98$; Wilcoxon matched-pairs signed rank test). In addition, there was no difference in the duration (Tactile-trials, $221 \pm 21$ ms; AudTac-trials, $241 \pm 15$ ms), onset (Tactile-trials, $350 \pm 34$ ms; AudTac, $406 \pm 38$ ms) or variance (Tactile-trials, $3.49 \pm 0.59$ ΔF/F; AudTac-trials, $3.87 \pm 0.64$ ΔF/F) of $Ca^{2+}$ transients in tuft dendrites during Hit (Tactile and AudTac) trials. These dendritic $Ca^{2+}$ events require NMDA receptor activation as they were abolished in 86% of dendrites by local application of the NMDA channel blocker APV ($n = 18/21$ dendrites, 3 mice). During correct rejections (Auditory-trials) (Fig. 5g), similar to the naive state (Fig. 1), auditory input alone evoked significantly larger $Ca^{2+}$ transients in tuft dendrites compared to tactile-trials (Fig. 5h; Auditory-trial, $3.26 \pm 0.30$ ΔF/F; Tactile-trial, $2.61 \pm 0.32$ ΔF/F, $n = 58$ dendrites, 6 mice; $p = 0.018$), however, these events had a significantly lower evoked rate ($7.5 \pm 0.8\%$ trials; $p = 0.0006$).

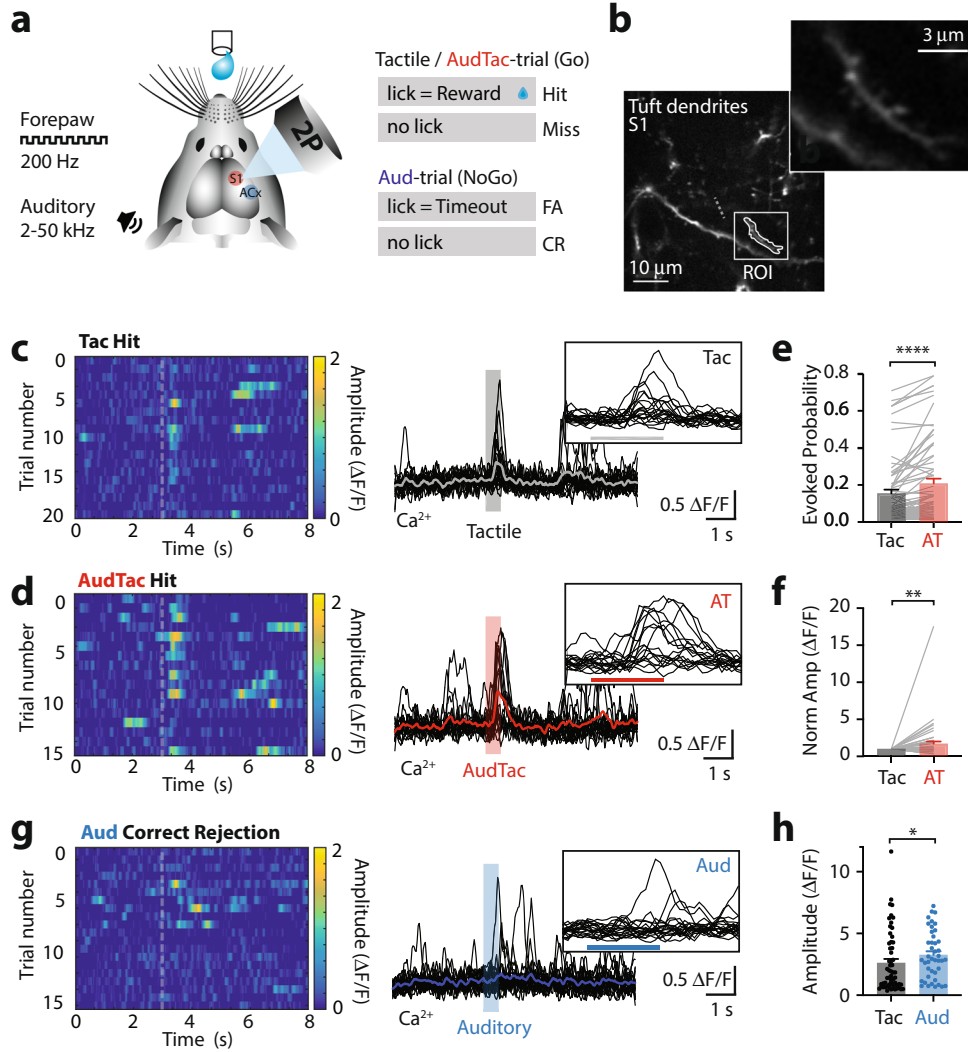

**Fig. 5 Auditory input during tactile goal-directed behavior enhances tuft Ca$^{2+}$ activity. a** Schematic of tactile-based goal-directed behavior paradigm. Mice received a water reward if they licked in response to tactile stimulus alone (Tactile-trial; 200 Hz, 500 ms) and paired tactile and auditory stimulus (AudTac-trial). On random trials, mice were also presented with auditory stimulus alone (Auditory-trial; NoGo, 2–50 kHz; 500 ms) which was not rewarded (Correct Rejection, CR) and a time out was given if mice licked (False Alarm, FA). **b** Example field of view. Ca$^{2+}$ activity from the tuft dendrites of L2/3 pyramidal neurons within the primary somatosensory cortex (S1) was recorded during task performance. Inset, zoom of dendrite from boxed region. **c** (left) Color heatmap of Ca$^{2+}$ signals from an example tuft dendrite during correct HIT performance in Tactile-trials. (right) Overlay of Ca$^{2+}$ traces during Tactile-trials. Inset, zoom of evoked Ca$^{2+}$ responses. **d** (left) Color heatmap of Ca$^{2+}$ signals from an example tuft dendrite during correct HIT performance in AudTac-trials. (right) Overlay of Ca$^{2+}$ traces during AudTac-trials. Inset, zoom of evoked Ca$^{2+}$ responses. **e** Percentage of trials with evoked Ca$^{2+}$ activity during Tactile-trials (black) and AudTac-trials (red). $n = 75$ dendrites, 6 mice; 39 trials$_{av}$; $p < 0.0001$; $p_{shuffled} = 0.850$; two-tailed Wilcoxon matched-pairs signed rank test. **f** Average peak amplitude of Ca$^{2+}$ responses during AudTac-trials (red) normalized to Tactile-trials (black). $n = 58$ dendrites, 6 mice; $p = 0.0025$; two-tailed Wilcoxon matched-pairs signed rank test. Only dendrites with evoked Ca$^{2+}$ transients during both Tactile- and AudTac- trials are included in the analysis. **g** (left) Color heatmap and (right) overlay of Ca$^{2+}$ signals during Auditory-trials from the example tuft dendrite in (**a**) and (**b**). Auditory-trials are not rewarded. Inset, zoom of evoked Ca$^{2+}$ responses. **h** Average peak amplitude of Ca$^{2+}$ responses during Tactile-trials (black) and Auditory-trials (blue). $n = 58$ dendrites, 6 mice; $p = 0.0025$; two-tailed Wilcoxon matched-pairs signed rank test. Error bars represent S.E.M. *$p < 0.05$, **$p < 0.01$, ****$p < 0.0001$.

Taken together, these results illustrate that dendritic Ca$^{2+}$ activity is enhanced when auditory input is paired with somatosensory input in a tactile-based goal-directed task.

**Auditory stimuli decreases the reaction time during tactile goal-directed behavior.** The presentation of auditory stimuli during this tactile-based goal-directed behavior did not influence task-related attention or arousal state as determined by pupillometry (Fig. 6a), with the relative change in pupil diameter not significantly different during correct performance in Tactile-trials

(0.130 ± 0.02 mm) and AudTac-trials (Fig. 6b; 0.122 ± 0.02 mm; $n = 6$ mice; $p = 0.22$). Pupil responses during Tactile-trials and AudTac-trials were both significantly larger than those during Auditory-trials (0.03 ± 0.01 mm; $n = 6$ mice; $p < 0.0001$), further illustrating that the auditory stimulus alone does not alter arousal state during this goal-directed task. To test whether auditory input during Tactile-trials alters behavior, we analyzed lick responses during correct performance (Hit trials) in expert mice (Fig. 6c). On average, expert mice reported the detection of the tactile stimulus by licking the port with an overall lick latency of 460 ± 33 ms ($n = 18$ mice). During the same session, when the

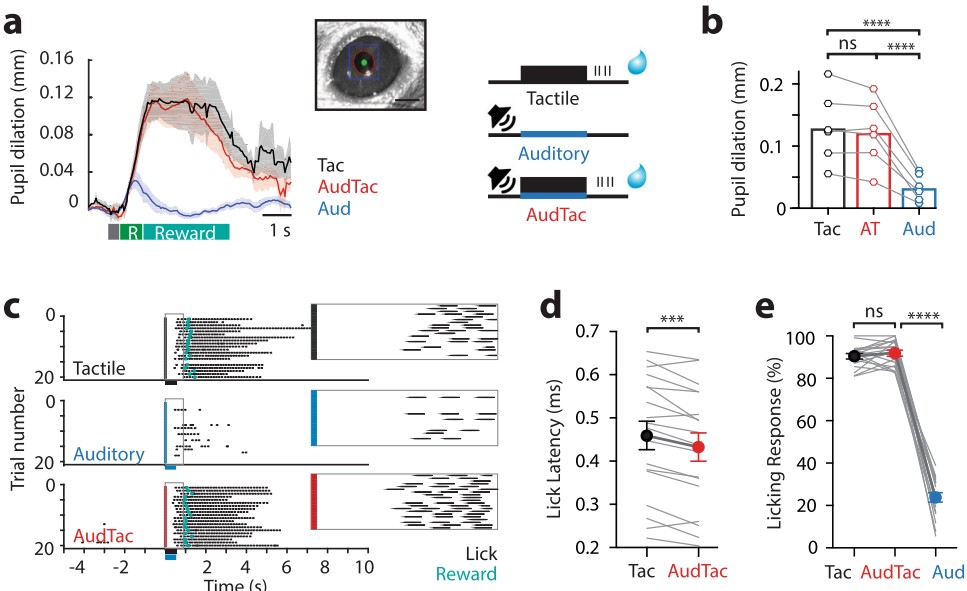

**Fig. 6 Auditory input during tactile goal-directed behavior decreases reaction time. a** Tactile goal-directed task: Mice were trained to receive a water reward if they licked a reward port in response to tactile stimulus alone (Tactile-trial; 200 Hz, 500 ms) and paired tactile and auditory stimulus (AudTac-trial). On random trials, mice were also presented with auditory stimulus alone (Auditory-trial; 2–50 kHz) which was not rewarded. Pupil diameter was measured during task performance illustrating Tactile-trials (black) and AudTac-trials (red) had similar dilation. Gray, stimulus; Green, response (R); Blue, reward. Inset, eye pupil ROI. **b** Pupil dilation during Tactile-trials was not significantly different to AudTac-trials (AT; $n = 6$ mice; 17 trials$_{av}$; $p = 0.31$ $p = 0.22$; $p_{shuffled} = 0.563$; Wilcoxon matched-pairs signed rank test). However, pupil dilation during both Tactile-trials and AudTac-trials was significantly greater than Auditory-trials ($n = 6$ mice; 10 trials$_{av}$; $p = 0.031$; $p_{shuffled} = 0.156$; Wilcoxon matched-pairs signed rank test). **c** Licking activity in an example mouse performing the goal-directed task during Tactile-trials (black), AudTac-trials (red) and Auditory-trials (blue). Dots, detected licks on lick port. Reward delivery, blue. Inset, expanded view of gray boxed region highlighting licking response from stimulus onset (colored line). **d** The licking response in AudTac-trials was faster than Tactile-trials ($n = 18$ mice; 50/61 trials$_{av}$; $p = 0.0002$; $p_{shuffled} = 0.610$; Wilcoxon matched-pairs signed rank test). **e** Percentage of trials with a lick in the response window during Tactile-trials (black), AudTac-trials (red) and Auditory-trials (blue). There was no significant difference in performance between Tactile-trials and AudTac-trials ($p = 0.327$; $p_{shuffled} = 0.229$; $n = 18$ mice; 61/50 trials$_{av}$; two-tailed Wilcoxon matched-pairs signed rank test). Error bars represent S.E.M. ***$p < 0.001$, ****$p < 0.0001$.

auditory stimulus was paired with a tactile stimulus (AudTac-trails), the lick latency was significantly reduced (Fig. 6d; 433 ± 33 ms; $n = 18$ mice; $p = 0.0002$). This decrease in response latency was not due to session differences, as mice reliably and robustly performed the task with only a 4.5 ± 2.5 ms difference in response latency between consecutive sessions (Tactile-trials, $n = 18$ mice). Furthermore, there was no correlation between the Tactile-trial reaction time and the decrease in response latency (Supplementary Fig. 12; $R^2 = 0.14$). Despite decreasing reaction time, there was no significant difference in the percentage of correct Tactile-trials (90.49 ± 1.27%) and AudTac-trials (Fig. 6e; 91.94 ± 1.42%; $n = 18$ mice; $p = 0.327$).

**Photoinhibition of axonal input in S1 abolishes the decrease in reaction time during tactile goal-directed behavior.** Is this enhanced behavioral response in the tactile-based goal-directed task due to axonal projections from the auditory cortex to fore-paw S1? To test this, we silenced these auditory axonal projections in forepaw S1 during the goal-directed behavior. To achieve this, mice were transfected with the inhibitory opsin, Archae-rhodopsin, in the auditory cortex (Fig. 7a and Supplementary Fig. 13). Following expression, axonal projections from the auditory cortex were photo-inhibited by a LED (590 nm) focused on the surface of forepaw S1 while mice performed the goal-directed task (Fig. 7b). Axonal photoinhibition abolished the decrease in response latency observed when auditory input was paired with tactile input (Fig. 7c; Tactile-trial, 417 ± 38 ms vs AudTac-trial, 408 ± 41 ms; $n = 6$ mice; $p = 0.563$). Light-activation alone did not alter lick latency, as the decrease in

reaction time observed during AudTac-trials was retained in sham mice injected with a control fluorophore (mus-eYFP; $p = 0.0312$; $n = 6$ mice) and in mice where light penetration into the cortex was blocked (Supplementary Fig. 14; $p = 0.03$; $n = 6$ mice;). Similar to control animals, input from the auditory cortex to forepaw S1 did not alter the percentage of correct responses, as mice performed at near perfect performance during photo-inhibition of auditory input (Fig. 7d and Supplementary Fig. 14; LED ON, 91.54 ± 3.44%; LED OFF, 99.01 ± 0.62; $p = 0.44$; $n = 6$ mice). These findings indicate that inhibiting the direct axonal projection from the auditory cortex to forepaw S1 abolishes the capacity of auditory input to enhance somatosensory encoding during tactile goal-directed behavior.

## Discussion

The function of the primary sensory areas of the cortex in information processing is unclear. Historically, primary sensory areas were thought to play an important role in the processing and encoding of sensory information necessary for perception of a single sense[35–37]. This, coupled with early work showing a lack of evidence for direct projections between primary sensory areas[38], supported the view that primary cortical areas were involved solely in unimodal stimulus processing. Here, we illustrate that the primary somatosensory cortex plays a more complex role in the processing of information, and combines sensory information from more than one modality to directly impact sensory-based behavior. Specifically, the auditory cortex sends direct projections to the primary somatosensory cortex, resulting in auditory-evoked activity in tuft dendrites of L2/3 pyramidal

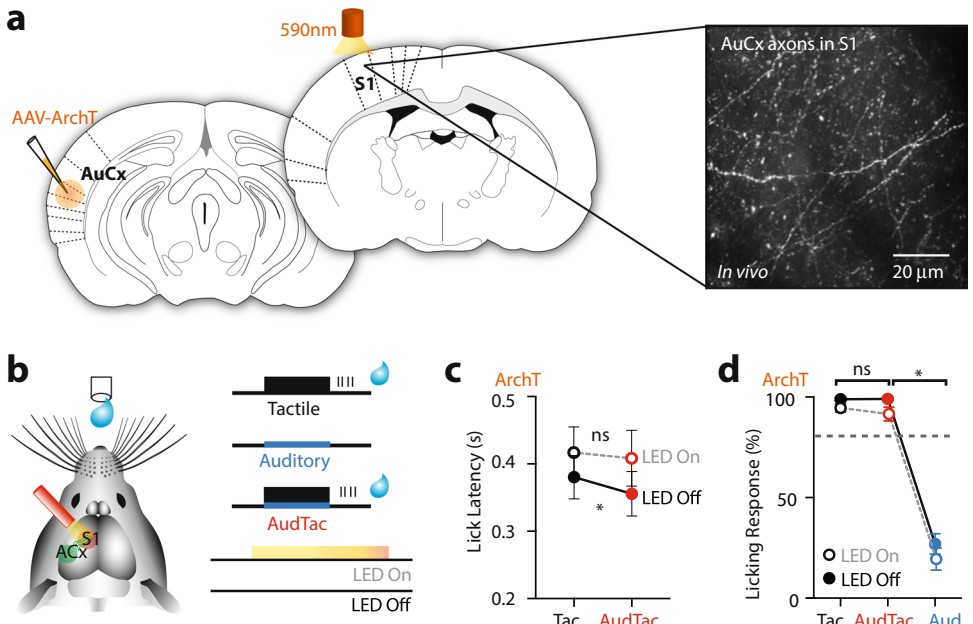

**Fig. 7 Photo-inhibiting the direct axonal projection from the auditory cortex to forepaw S1 abolishes the capacity of auditory input to decrease the behavioral response latency during tactile goal-directed behavior. a** The inhibitory opsin, Archaerhodopsin (ArchT), was injected into the auditory cortex and axonal projections in forepaw S1 were photo-inhibited (590 nm) during the goal-directed behavior. Example image of auditory-cortex axons transfected with Archaerhodopsin in forepaw S1 (~50 μm below pia). **b** Schematic of experimental paradigm. Mice previously injected with Archaerhodopsin were trained to receive a water reward if they licked a reward port in response to tactile stimulus alone (Tactile-trial; 200 Hz, 500 ms) and paired tactile and auditory stimulus (AudTac-trial). On random trials, mice were also presented with auditory stimulus alone (Auditory-trial; 2–50 kHz) which was not rewarded. LED (590 nm) was either focused on the surface of forepaw S1 (LED On) or not (LED Off) while the mouse performed the goal-directed task. **c** The lick latency during Tactile-trials and AudTac-trials were not significantly different during photoinhibition of auditory axons in forepaw S1 (LED On; $n = 6$ mice; 37/33 trials$_{av}$; $p = 0.563$; $p_{shuffled} = 0.688$; Wilcoxon matched-pairs signed rank test). **d** Percentage of trials with a lick in the response window during LED On (orange) and LED Off (black). There was no significant difference in performance during LED On ($n = 6$ mice; 37/33 trials$_{av}$; $p = 0.44$; $p_{shuffled} = 0.844$; two-tailed Wilcoxon matched-pairs signed rank test). Error bars represent S.E.M. $*p < 0.05$, $***p < 0.001$, $****p < 0.0001$.

neurons in S1. When presented with tactile stimulus, this monosynaptic auditory input enhanced the encoding of the tactile stimuli by increasing the $Ca^{2+}$ response in tuft dendrites as well as the number of somatic action potentials. Furthermore, we show that auditory input impacts tactile-based behavior, decreasing the reaction time during a goal-directed task. Photo-inhibition of axonal projections from the auditory cortex to forepaw S1 abolished the impact of auditory input on reaction speed, illustrating the importance of this input during goal-directed behavior.

We find that auditory input enhances sensory encoding in L2/3 pyramidal neurons in forepaw S1. Recent work indicates a similar excitatory impact of auditory input in barrel cortex[39]. However, in contrast, response suppression to auditory input[13] and multiple unimodal stimuli[40, 41] has been demonstrated in other primary cortical areas and cell types. These contrasting results illustrate the complexity of multisensory integration, which targets different cell types as well as different cortical layers and regions, leading to both excitatory and inhibitory cell-specific changes in cellular processing. Indeed, our results highlighted the different influence of auditory input on sensory processing in L2/3 and L5 pyramidal neurons in forepaw S1. We find that combining monosynaptic input from the auditory cortex with tactile stimulation of the forepaw enhanced dendritic $Ca^{2+}$ responses and somatic action potential output in L2/3, but not L5, pyramidal neurons in S1. This may result from the increased electrotonic remoteness of L5 apical tuft dendrites from the soma compared to L2/3 pyramidal neurons[42, 43]. It must also be noted that the influence of auditory stimuli on sensory processing in L5 pyramidal neurons may be different during active behaving

conditions. Consistent with our findings, computational differences in sensory processing have been shown between L2/3 and L5 pyramidal neurons within the primary visual cortex during visuomotor mismatch[44] and in the barrel cortex during locomotion[45]. Selectivity of multisensory information processing has also been reported in auditory projections to the visual cortex[14], where orientation selectivity of L2/3, not layer 4, excitatory neurons was sharpened by the presence of sound. In contrast, the impact of visual information on coding in auditory cortex is almost exclusively found in deeper infragranular layers[46]. Our results, combined with previous findings, illustrate that principal neurons within primary cortical regions can process information differently.

Paradoxically, in contrast to the somatic response, auditory stimuli evoked larger $Ca^{2+}$ transients in tuft dendrites of L2/3 pyramidal neurons in forepaw S1 than tactile stimulation. This may be from differences in the synaptic location of these two different sensory inputs onto pyramidal neurons. In the cortex, bottom-up sensory information typically targets the middle cortical layers, whereas top-down information usually courses through the upper cortical layers where the distal dendrites of pyramidal neurons stratify[37, 47, 48]. Using fluorescence imaging, we illustrated that auditory input is predominately restricted to upper cortical layers. Therefore, auditory input would primarily synapse onto tuft dendrites in S1, whereas tactile somatosensory input would primarily target basal dendrites[49]. Using the Allen Brain Atlas, we found a similar pattern of projections from the auditory cortex in both forelimb and the hindlimb regions of S1, whereas the barrel cortex receives projections that are more widespread across the cortical layers. Therefore, in different

sensory cortices, and even within different regions of S1, auditory axonal projections may target different dendritic regions or cell types, and may therefore have different influences on sensory processing. Furthermore, our findings illustrate that low auditory intensities have the greatest effect on encoding of the tactile stimulus in forepaw S1. Although the cause of this surprising result remains to be tested, it may be a consequence of the encoding of sound in the auditory cortex being heavily modulated by inhibition[50]. Although out of the scope of this study, this finding will lead to exciting new avenues of research. In contrast to $Ca^{2+}$ transients in tuft dendrites, the soma of L2/3 pyramidal neurons fired more action potentials during tactile stimulation compared with auditory stimulation alone. Again, this is likely to be a consequence of the location of synaptic input, as proximal tactile input onto basal dendrites is electrotonically closer to the soma than distal input from the auditory cortex. Additionally, the relative influence of different sensory modalities on neural activity may be dependent on the specific training paradigm[51, 52] and stimuli used[2, 53]. By testing the influence of a range of different auditory and tactile stimuli on tactile-encoding, we illustrate that the evoked rate, but not amplitude, of the multisensory integration in tuft dendrites is influenced by stimulus characteristics. Therefore, multisensory integration is a dynamic process which would alter according to the specific sensory environment.

Auditory input specifically enhances sensory processing in L2/3 pyramidal neurons, which target both local neurons[28–30] and neurons in distant cortical and subcortical areas[31, 32]. Despite auditory input causing a significant increase in the tactile-evoked firing in L2/3 pyramidal neurons, there was no measurable influence of auditory input on the subthreshold response at the soma in vivo. Although there was a subthreshold voltage response recorded during photoactivation of auditory inputs in vitro, this could be because photoactivation of axons from the auditory cortex in vitro recruits many more inputs than auditory stimuli in vivo and does so in a highly synchronized way. The disconnect between subthreshold responses and enhanced action potential output in vivo has been reported previously in hindpaw S1[34] and is likely to be due to the generation of dendritic spikes in the distal apical dendrites, which can have a direct impact on action potential generation[16, 19, 24, 54, 55]. Consistent with this idea, auditory input increased tactile-evoked $Ca^{2+}$ transients in distal tuft dendrites of L2/3 pyramidal neurons. This may result from the active integration of synaptic input from the two different pathways, or possibly from passive boosting of the tactile-evoked $Ca^{2+}$ transient by auditory input. Since auditory input also increased the evoked rate of $Ca^{2+}$ transients during tactile-based behavior, this suggests auditory input actively integrates with tactile input to generate a new detectable event in vivo. Taking these results together, they suggest a non-linear interaction between top-down intra-cortical and bottom-up sensory inputs. This connectivity strategy provides the entire cortex with an intermediate stage of processing which, due to the integration of internal feedback, may act to lower the threshold for perception[56]. Indeed, it is possible that pairing an auditory stimulus with a threshold-level tactile stimulus would have an effect on the behavioral performance by reducing the perceptual detection threshold, however, this remains unaddressed and should be explored by further studies.

During tactile goal-directed behavior, auditory stimuli decreased reaction time by ~30 ms. Taking into account a minimum reaction time in sensory decision-based behaviors of ~200 ms[57], the small effect of auditory input on the overall behavioral response equates to a more substantial portion of the variable response time (~15%). Enhanced reaction times may have a profound effect on behavior in a multisensory environment, and even small changes in reaction time may equate to large changes in sensory perception, decision making,

and overall survival. To assess whether this decrease in reaction time was due to monosynaptic auditory input, we photo-inhibited the axonal projections from the auditory cortex to forepaw S1. Photo-inhibition of auditory axons abolished the impact of auditory input on reaction speed, suggesting that the direct auditory input to forepaw S1 acts to enhance the reaction time during goal-directed behavior. While long-duration (second) activation of ArchT can have excitatory effects[58], activation of ArchT for shorter durations, as used in our study, has previously been shown to be inhibitory[59]. Modulation of reaction times during photoinhibition of the primary somatosensory cortex have also been observed in a texture discrimination task, however, in this case reaction time was driven by the recruitment of cortical inhibition by L5 pyramidal neurons[60].

Integrating information from multiple senses is crucial to survival. The cortex is central to this process, connecting information from different senses in a layer and neuron-specific manner. Here, we demonstrate that this can occur at the level of single neurons in the primary somatosensory cortex, influencing the sensory encoding in tuft dendrites of pyramidal neurons. This localized cellular enhancement of neural activity endows the cortex with multiple independent integrative units which may facilitate the coordination of activity across different cortical areas during sensory perception[8] and act to increase cortical computational power[61]. Overall, our study adds to an increasing body of literature demonstrating that primary cortical areas process information from multiple senses. These studies, combined with our findings, challenge the classic view of information processing in the cortex as being mainly hierarchical and segregated.

## Methods

All experiments were conducted in strict accordance with the Code of Practice for the Care and Use of Animals for Scientific Purposes (National Health and Medical Research Council, Australia). The guidelines were given by the veterinary office and approved by the Animal Ethics Committees of The Florey Institute of Neuroscience and Mental Health, University of Melbourne, and of the Australian National University.

**Virus injections.** Mice (C57BL/6; P30–P55) were anaesthetized with isoflurane (1–3% in 0.75 L/min $O_2$) and body temperature was maintained at 36–37 °C. Eye ointment was applied to prevent dehydration and meloxicam (1–3 mg/kg, Ilium) was intraperitoneally injected for anti-inflammatory action. The skin was disinfected with ethanol 70% and betadine, and a small slit was made in the skin to expose the skull. A small craniotomy (0.7 × 0.7 mm) was then made over the brain region of interest (ROI) and the dura was left intact. For labelling the auditory cortex with channelrhodopsin (ChR2; AAV1.hSyn.ChR2(H134R)-eYFP.WPRE. hGH), 100 nl was injected at 2.5 mm posterior to bregma and 4.5 mm lateral from midline (at a cortical depth from pia of 200–500 μm). For sparsely labelling the primary somatosensory cortex (forepaw) with a $Ca^{2+}$ indicator, a mix of Cre-dependent genetic $Ca^{2+}$ indicator GCaMP6f (AAV1.Syn.Flex.GCaMP6f.WPRE. SV40) and diluted Cre (1:6000; AAV1.hSyn.Cre.WPRE.hGH) was injected into either L2/3 (at a depth of 450 μm) or L5 (at a depth of 700 μm) at the stereotaxic coordinates, 0 mm from bregma and 2 mm from midline. For densely labelling the primary somatosensory cortex (forepaw) with a $Ca^{2+}$ indicator (for population imaging of L2/3 pyramidal neuron somata), we injected the genetic $Ca^{2+}$ indicator GCaMP6f (AAV1.Syn.GCaMP6f.WPRE.SV40; 100 nl) into the primary somatosensory cortex (as above). Layer specificity of the injection site and identification of the recorded cell-type was confirmed after each experiment by visualizing the entire dendritic arbor and location of the associated somata. After slowly retracting the microcapillary pipette, the skin was sutured and the mouse was able to recover for at least 3 days prior to any further experimental procedures.

**Head-post implantation and cranial window surgery.** To surgically implant the head-post for recordings in the awake state, mice were anaesthetized with isoflurane (1–3% in 0.75 L/min $O_2$) and intraperitonally injected with Meloxicam (1–3 mg/kg, Ilium). Throughout the surgery, body temperature was maintained at 36–37 °C. Lidocaine (20 mg/ml, Ilium) was topically injected around the surgical site before the skin was cut to expose the skull. A custom-made metal head-bar was then attached to the skull using dental cement (C&B Metabond®, Parkell). For electrophysiological recordings in the awake state, the remaining exposed skull was covered with transparent dental cement and the mouse was returned to their home cage for at least 3 days before habituation commenced. For $Ca^{2+}$ imaging experiments, a craniotomy was performed (3 mm diameter) over the virus injection site in the primary somatosensory cortex. A circular coverslip (3 mm diameter, size

#1) was placed over the craniotomy, and sealed with glue and dental cement before the entire surface was covered with inert silicon (kwik-cast, WPI). Mice were returned to their home cage for at least 3 days before behavioral training commenced. In one cohort of mice, a double window was used by putting together two coverslip of different sizes (3–3.5 mm) using UV curable glue. This helps to reduce motion artifacts and stabilize the cranial window implant.

**EMG recordings**. In a subset of experiments, two electrodes were inserted in the neck muscles bilaterally during head-post surgery for nuchal EMG recordings. The EMG signal was recorded using differential amplifier DP-301 (Warner Instruments), band-pass filtered between 300 and 10,000 Hz, digitized and recorded at 20 kHz using custom-written Igor Pro (Wavemetrics) software.

**Pupil tracking**. In some experiments, the pupil of the mouse was monitored during passive stimulus presentation or task performance in the expert mouse, to measure attention level and arousal state. The pupil dilation was tracked using an high speed CMOS camera (Blaser aCA1300) mounted on a 50 mm lens (Kowa 50 mm/F2.8) and analysis was performed using custom software provided by Viktor Bahr, Jens Kremkow and Robert Sanchez from Charité University Berlin. The traces were extracted and baseline subtracted. The relative change in pupil diameter during stimulus presentation or task performance was then measured as the maximum value in a window of 1 s following stimulus onset (once the trace crossed 2 standard deviations from baseline activity). Values are reported as mean changes in pupil size (mm) per mouse.

**Sensory stimulus**. Tactile stimulus was delivered to the contralateral forepaw (200 Hz; 500 ms) via a piezo-electric buzzer (Microdrive). Auditory stimulation was evoked using a broadband noise stimulus (2–50 kHz, 500 ms) played through a speaker (Logitech) placed ~5–10 cm from the contralateral ear. Both tactile and auditory stimuli were above perception threshold, as determined by responses during pupil tracking. In a subset of experiments, the duration of the tactile stimuli (75, 250, and 500 ms) and the intensity of the broadband noise stimulus (50, 70, and 90 dB) was manipulated to test the effect of different stimuli parameters on multisensory integration. All experiments were performed within a sound-proofed faraday cage with a white noise ambient sound constantly played to further isolate and exclude any external sound cue (not relevant to the task). The tactile stimulus produces sound measured at 1–2 dB, which is considerably less than the auditory stimulus and background ambient sound. Both tactile and auditory stimuli were generated and delivered using Arduino micro-processing boards (Arduino Uno) and Bpod (Sanworks) with custom-written MATLAB (MathWorks) software.

**Whole-cell recordings in the awake state**. Mice (C57BL/6; P25–39) which had previously had a head-post implanted (>5 days prior) were gradually habituated to head restriction for 4–9 days. Here, mice were fixed to the recording frame and their paws rested unaided on either an active (contralateral) or inactive (ipsilateral) vibrating button. To prevent auditory startle response, the speaker delivering the auditory stimulation was located at a distance >30 cm from the mouse. During habituation, experimental stimuli (auditory broadband noise stimulus (2–50 kHz, 50–60 dB, 1 s); tactile (200 Hz, 1 s) or both together) were presented randomly every 10–20 s for at least 3 sessions (30–60 min each). Once habituated, a craniotomy (1 × 1 mm) was performed over the forepaw area of the primary somatosensory cortex (AP, 0 mm; ML, +2.2 mm) under isoflurane anesthesia (1–3% in 0.75 L/min O$_2$). The brain was covered with agar and then inert silicon (kwik-cast, WPI). Animals were allowed to recover for at least 2 h before whole-cell patch-clamp experiments were performed for a maximum of 2 h over 2 consecutive days. During a recording session, mice were placed on the head-fix frame and their paws naturally rested on a vibrating button. The silicon protective cover was removed, and normal ringer (in mM; 135 NaCl, 5.4 KCl, 1.8 CaCl2, 1 MgCl2, 5 HEPES) was used to bathe the craniotomy throughout the experiment. Whole-cell in vivo patch-clamp recordings were obtained from either L2/3 (~200 μm below pia) or L5 (~600 μm below pia) pyramidal neurons using a patch pipette (resistance 4–6 MΩ) filled with an intracellular solution containing (in mM) 115 potassium gluconate, 20 KCl, 10 mM sodium phosphocreatine, 10 HEPES, 4 Mg-ATP, 0.3 Na-GTP, adjusted to pH 7.3–7.4 with KOH. The patch pipette was inserted into the brain at an angle of 45° relative to the cortical surface, to a depth of ~200 μm. The pipette was then advanced in steps of 1 μm (for a maximum distance of 200 μm in the hypotenuse trajectory) until a neuron was encountered. Whole-cell voltage recordings were performed from the soma using Dagan BVC-700A amplifiers and sampled at 20 kHz. Custom-written Igor Pro (Wavemetrics) software was used for both acquisition and analysis and no correction was made for the junction potential. The identity of the recorded pyramidal cell in in vivo blind recordings was confirmed using the recording depth and voltage response to current steps. Once a whole-cell recording was obtained, the voltage response to current steps (50 pA; 800 ms) was recorded to characterize the neuron. Mice were then exposed to auditory broadband noise stimulus (2–50 kHz, 60 dB, 1 s) and/or tactile stimulus (200 Hz, 1 s) at inter-trial intervals of 10 s. In a subset of neurons which had a low rate of action potential firing, positive holding current was applied to the neuron via the patch pipette (~50 pA) to provide additional depolarization to lower the threshold for action potential generation. Only recordings where greater than 14

trials of each stimulus was presented to the mouse were included in the analysis. Where reported, neurons were filled with fluorescent biocytin for posthoc cell identification and morphological reconstruction using online software, NeuTube.

**Whole-cell recordings in the anaesthetized state**. Mice (C57BL/6; P42–63) were initially anaesthetized with isoflurane (1–3% in 0.75 L/min O$_2$) before urethane anesthesia (intraperitoneal, 1.6 g/kg, Sigma) was administered. Anesthesia was monitored throughout the experiment, and a top-up dose of 10% of the initial urethane dose was administered when necessary. Body temperature was maintained at 36–37 °C. Lidocaine (20 mg/ml, Ilium) was injected around the surgical site on the scalp and the head was stabilized in a stereotaxic frame by a head-plate attached to the skull with dental cement (paladur, Heraeus). A craniotomy was performed over the forepaw area of the primary somatosensory cortex, S1 (~1.5 × 1.5 mm$^2$), centered at bregma and 2.2 mm lateral from midline. The dura was surgically removed and normal rat ringer (as above) was used to bathe the craniotomy throughout the experiment. Whole-cell in vivo patch-clamp recordings were performed using a patch pipette (resistance 6–9 MΩ) filled with intracellular solution (as above). The patch pipette was inserted into the brain at an angle of 30° relative to the cortical surface, to a depth of ~200 μm (to target L2/3 pyramidal neurons). The pipette was then advanced in steps of 1 μm (for a maximum distance of 200 μm in the hypotenuse trajectory) until a neuron was encountered. Whole-cell voltage recordings were performed from the soma using Dagan BVC-700A amplifiers and were filtered at 10 kHz. Once a whole-cell recording was obtained, the voltage response to current steps (50 pA; 800 ms) was recorded to characterize the neuron. In a subset of neurons which had a low rate of action potential firing, positive holding current was applied to the neuron via the patch pipette (~50 pA). Custom-written Igor Pro (Wavemetrics) software was used for both acquisition and analysis and no correction was made for the junction potential.

For post-hoc identification and partial reconstruction of the recorded neurons, the cellular tracer, 5-(and-6)-Tetramethylrhodamine Biocytin (0.01–0.02%) was included in the intracellular solution in a subset of experiments. For visualization, brains were sliced via a vibratome into 100 μm sections and fluorescent neurons were visualised using a confocal microscope (561 nm excitation, 566–669 nm bandpass emission filter).

**Whole-cell in vitro recordings**. Mice (P30–P35) previously injected with ChR2 in the auditory cortex (>14 days prior) were anaesthetized with isoflurane (3–5% in 0.75 L/min O$_2$) before decapitation. The brain was then rapidly transferred to ice-cold, oxygenated cutting solution containing (in mM): 110 Choline Chloride, 11.60 Na-ascorbate, 7 MgCl$_2$, 3.1 Na-pyruvate, 2.5 NaH$_2$PO$_4$, 0.5 CaCl$_2$, 10 Glucose and 26 NaHCO$_3$. Coronal slices of the primary somatosensory cortex (300 μm thick) were cut with a vibrating microslicer (Leica Vibratome 1000 S) and incubated in an incubating solution containing (in mM): 92 NaCl, 2.5 KCl, 1.2 NaH2PO$_4$, 30 NaHCO$_3$, 3 Na-pyruvate, 2 CaCl$_2$, 2 MgCl$_2$ and 25 Glucose at 35 °C for 30 min, followed by incubation at room temperature for at least 30 min before recording. All solutions were continuously bubbled with 95%O$_2$/5%CO$_2$ (Carbogen). Whole-cell patch-clamp somatic recordings were made from visually identified pyramidal neurons using DIC imaging and a CCD camera (PL-B957U, Pixelink). During recording, slices were constantly perfused at ~2 ml/min with carbogen-bubbled artificial cerebral spinal fluid (ACSF) containing (in mM): 125 NaCl, 25 NaHCO$_3$, 20 HEPES, 3 KCl, 1.25 NaH$_2$PO$_4$, 2 CaCl$_2$, 1 MgCl$_2$, and 25 Glucose maintained at 30–34 °C. Patch pipettes were pulled from borosilicate glass and had open tip resistance of 5–7 MΩ filled with an intracellular solution containing (in mM):130 potassium gluconate, 10 KCl, 10 sodium phosphocreatine, 10 HEPES, 4 Mg-ATP, 0.3 Na$_2$-GTP, and 0.3% biocytin adjusted to pH 7.25 with KOH. All recordings were made in current-clamp using a BVC-700A amplifier (Dagan Instruments, USA). To ensure direct comparison, recordings were made from L2/3 and L5 pyramidal neurons from the same slice.

**Habituation and behavioral training for Ca$^{2+}$ imaging during behavior**. Mice which had previously undergone head-bar implantation surgery were water restricted (5–2 cycle, 1 ml/day on restriction days; ad libitum access to food). After >2 days of water restriction, mice were gradually habituated to head fixation and the microscope setup for 4–9 days. Once habituated, behavioral training commenced. Here, mice were head-fixed to the recording frame and their paws rested unaided on either an active (contralateral) or inactive (ipsilateral) piezo-electric buzzer (Microdrive). Tactile stimulus was delivered to the contralateral forepaw (200 Hz; 500 ms). Auditory stimulation was evoked using a broadband noise stimulus (2–50 kHz, 75 dB, 500 ms) played through a speaker (Logitech) placed ~5–10 cm away from the contralateral ear. W Ambient white noise was constantly played during the training session to isolate and exclude from any external sound cue (not relevant to the task). Both tactile and auditory stimuli were generated and delivered using Arduino micro-processing boards (Arduino Uno) and custom-written MATLAB (MathWorks) software. Behavioral training was performed in a systematic manner. (1) The association phase of training involved training the mouse to associate the presentation of the tactile forepaw stimulation with an automatically presented sucrose reward (10 μl, 10% sucrose in water). (2) Next, the mouse learnt to lick the reward spout for reward delivery. Here, sucrose water

reward was only delivered if the mouse licked the reward spout within a 1–2 s period following delivery of the tactile stimulus. After the inter-trial interval (2–3 s), trial initiation was triggered only when the mouse was not spontaneously licking for at least 1–2 s. Hit rates were calculated as the number of correct trials where the animal licked within the response period, divided by the total number of trials in a session. Once a hit rate of 80% was achieved consistently over a period of 3 consecutive days, animals progressed to the next phase. (3) The auditory stimulus alone was presented and the mouse learnt to ignore it and withhold licking while continuing to report the detection of the tactile stimulus. Licking after broadband auditory noise stimulus alone incurred a timeout punishment (2 s time-out which was re-triggered if licks were detected). Percentage of correct trials for this phase were calculated as the sum of hit (trials with correct tactile detection, as the previous phase) and correct rejections (trials where the mouse did not lick after presentation of the auditory stimulus) divided by the total number of trials in a session. Also here, once a hit rate of 80% was achieved consistently over a period of 3 consecutive days, animals were moved on to the test phase. (4) The testing phase consisted of the delivery of the following stimuli: tactile alone stimulus; auditory alone stimulus; tactile plus broadband noise auditory stimulus with a 0 ms temporal offset. Lick responses after tactile stimulus (either alone or paired with auditory stimulus) was rewarded with a 10 µl drop of sucrose water. Licking after broadband auditory noise stimulus alone incurred a timeout punishment (2 s time-out which was re-triggered if licks were detected). This behavior was deemed to be goal-directed as it was extinguished within a single session where licking responses were not water rewarded. Behavioral training and testing protocols were custom-written and presented using BPod (Sanworks), and MATLAB (MathWorks) was used to collect data.

**Two-photon Ca²⁺ imaging**. Two-photon imaging was performed through the cranial window implanted in mice previously transfected with the Ca²⁺ indicator GCaMP6f (either dense or sparsely labelled—see viral injections section). In either expert or naive mice, Ca²⁺ transients were recorded from tuft dendrites or soma of pyramidal neurons within the primary somatosensory cortex. For imaging of dendrites and soma from L2/3 pyramidal neurons within the same neural population, the focal plane was changed from imaging in the upper layers (dendrites) or layer 2/3 (soma) throughout the experiment. GCaMP6f was excited at 940 nm (~30 mW at the back aperture) with a titanium sapphire laser (140 fs pulse width; SpectraPhysics MaiTai Deepsee) and imaged on a Sutter MoM through a 16x Nikon objective (0.8 NA). Emitted light was passed through a dichroic filter (565dcxr, Chroma Technology) and short-pass filtered (ET525/70-2p, Chroma Technology) before being detected by a GaAsP photomultiplier tube (Hamamatsu). Images were acquired at a frequency of 30 Hz (512 × 512 pixels) using ScanImage software (Vidrio Technologies). The images were motion-corrected with a custom-written Matlab script. All imaging from tuft dendrites was performed in higher-order branches beyond the bifurcation point. ROI was manually drawn on ImageJ and fluorescence signal extraction was performed with a custom MATLAB script. All trials with motion in the z-axis were excluded from the analysis.

**Optogenetic modulation of auditory inputs**. To specifically manipulate auditory projections into the primary somatosensory area (forepaw), 150 nl of the inhibitory opsin Archaerhodopsin (AAV1.CAG.ArchT.GFP.WPRE.SV40, Addgene pasmid #29777) was injected into the auditory cortex (750 µm below pia) of PN30 mice and placed a cranial window over the primary somatosensory cortex with the same procedures described in the above sections (virus injections and cranial window surgery). The control group was injected with a GFP (muGFP[62], following the same procedures. After mice reached expert level in the behavioral task, a 590 nm LED (15 mW) was placed over the window for photoinhibition of auditory inputs during behavior. The light was carefully adjusted so that the focal plane was positioned to optimally deliver the LED light into layer 1 of the primary somatosensory cortex. The design of the experiment consisted in a blocks of trials with no light (LED Off) and with LED On (100 each). In LED On trials the light was constantly delivered from 0.5 s before stimulus onset until the end of the trial (~5.50 s). To test the influence of LED on behavior, two controls were performed. (1) Mice were injected with muGFP, and underwent the same training and testing protocol as the Archaerhodopsin cohort. (2) In mice injected with Archae-rhodopsin, the cranial windows were temporary covered with silicon (kiwk-Cast, WPI) to prevent the light to reach the brain. Lick latencies were analysed in a window between 0.1 and 1 s from stimulus delivery with a custom MATLAB script. The injection sites and viral spread were checked post-hoc in mice injected with Archaerhodopsin to confirm expression was limited to the auditory cortex. (In vitro) photoactivation of axonal projections from the auditory cortex in the brain slice from the somatosensory cortex was achieved by briefly passing 470 nm LED light (10 ms) through a 60× Olympus objective onto the somatosensory cortical slice (10 trials per LED intensity). To test for monosynaptic callosal input, TTX (1 µM) and 4-AP (100 µM) were added to the bath ACSF.

**Drug application—in vivo and in vitro**. For in vivo block of NMDA receptors, the cranial window was removed under isoflurane anesthesia (3–5% in 0.75 L/min O₂) and APV was topically applied onto the brain surface (10 mM, Tocris). The glass coverslip was then immediately resealed to perform imaging. For in vitro

pharmacological manipulations, drugs were bath applied by inclusion in the circulating ACSF at the required concentration.

**Data analysis**. (Whole-cell recordings) Custom-written Igor Pro (Wavemetrics) software was used for the acquisition and analysis of whole-cell recordings. The latency of tactile response was determined by using the threshold calculated from baseline Vm (mean ± 2 sd). The analysis window for action potential reporting was 1 s following stimulus presentation for all recorded neurons. When represented as normalized, all responses were normalised to the response to 200 Hz tactile-only stimulus. (Behavior) Only mice that reached 80% expert performance in the tactile-trials, and had <40% false alarms during Auditory-trials were included in the analysis. (Ca²⁺ imaging) Custom-written MATLAB (MathWorks) software was used for analysis of Ca²⁺ imaging data. Ca²⁺ events were detected in a 1 s window from stimulus onset, when they crossed a threshold value of 3 standard deviation measured on the baseline (3 s, from trial onset to stim onset). The amplitude and duration of each calcium transient were measured as the peak and half width at half maximum of the event occurred during the detection window. N represents dendrites, mice. The probability of Ca²⁺ transients was measured as the number of events divided by the number of trials. Peak amplitudes were only measured on Ca²⁺ transients above threshold, whereas, evoked probability included trials with no events, resulting in the reporting of conditions without responses (where probability = 0). Therefore sample number for amplitude and probability of responses might differ. Reporting of normalized data is reported in the manuscript where evoked probability is normalized to the average tactile response (to prevent overrepresentation in a heterogeneously responding population). Ca²⁺ responses were smoothed using a Savitzky–Golay filter with a 2nd order polynomial and a 7-sample window and only transients longer than 250 ms were included in analysis. Unless differently specified, the amplitudes and durations are reported as the mean value of the detected events for each ROI. 3D reconstruction and tracing of tuft dendrites to identify the soma of origin was performed posthoc using NeuTube software. (Statistics) Measurements were taken from distinct samples and all numbers are indicated as mean ± SEM. When comparing two populations of data, significance was determined using two-sided nonparametric tests (paired: Wilcoxon matched-pairs signed rank test; unpaired: Mann–Whitney test) at a significance level of 0.05. When multiple populations of data were compared, nonparametric one-way ANOVA Friedman or Kruskall–Wallis with Dunn post hoc tests were used for paired and unpaired comparison, respectively. To calculate the shuffled p values, the data were randomly re-sampled from a pooled dataset containing all of the variables and new random comparison groups were created and compared. The number of trials tested for each condition is reported as the average number for all recordings in response to the stimulus.

**Reporting summary**. Further information on research design is available in the Nature Research Reporting Summary linked to this article.

## Data availability
The raw source data contributing to this study are available from the corresponding author on request. Source data are provided with this paper.

## Code availability
The analysis code contributing to this study are available from the corresponding author on request.

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

## Acknowledgements

We would like to thank members of the Palmer laboratory for helpful comments on the manuscript and Dan Scott for providing the AAV1/2-muGFP GFP. This work was supported by the NHMRC (APP1086082, L.M.P. and G.J.S.; APP1063533, L.M.P.), the Sylvia and Charles Viertel Charitable Foundation (L.M.P.) and the Australian Research Council Centre of Excellence for Integrative Brain Function (ARC Centre Grant CE140100007).

## Author contributions

L.G., D.A., and L.M.P. designed the experiments. D.A. and R.G. performed and analyzed the anaesthetized and awake in vivo patch-clamp recordings respectively. L.G. performed and analyzed the two-photon Ca$^{2+}$ imaging experiments. L.G. and D.A. performed and analyzed all behavioral experiments. S.H. performed and analyzed the in vitro patch-clamp recordings. L.G., D.A., G.J.S., and L.M.P. wrote the manuscript.

## Competing interests
The authors declare no competing interests.
