## [Peer Review File · Nature Communications]

REVIEWER COMMENTS

Reviewer #1 (Remarks to the Author):

Godenzini and colleagues investigated the cortical basis supporting the integration of multisensory information in the somatosensory cortex in mice.

The results add new insights to the body of work on multisensory integration. A striking finding is the selectivity of auditory inputs onto layer 2/3 but not layer 5 pyramidal neurons in the forepaw cortex. This distinction was demonstrated forcibly using in vitro and in vivo method. However, the authors did not quite fully explore the difference to show if the distinction is also behaviorally significant.

Notwithstanding this critique (which may be difficult to do for technical reasons), the study is solid. The experiments were performed carefully using an array of modern approaches. The results generate new understanding for how auditory inputs may be integrated in the somatosensory cortex of the mouse. The manuscript is well-written and logically organized.

Major comments:

- Figure 5 shows difference in dendritic calcium signals between Tactile-trials and AudTac-trials. While the authors intuit that this difference stems from the sensory component, it is possible that there may be contributions to these calcium responses from motor (licking) and/or outcome (sucrose-water reward) components. It is possible that the mouse may lick earlier and more vigorously during AudTac-trials, which would produce a motor-related difference. Additionally, a prior study showed substantial dendritic calcium responses in layer 2/3 pyramidal neurons in the barrel cortex in response to rewards (Lacefield et al., Cell Rep, 2019). For example, a contribution from reward may have an additive effect with combined auditory-tactile stimuli. If the authors can isolate the sensory-evoked calcium response from the reward component, they may see a greater enhancement for paired stimuli versus tactile stimuli. Can the contributions of sensory, motor, and outcome be dissociated? A linear regression analysis (e.g., Siniscalchi MJ et al., Cereb Cortex, 2019), with tactile stimulus, auditory stimulus, lick rate, and reward as factors, could be informative.

- Throughout the study, only one type of both auditory and tactile stimulus was used to show that the activation of one sense can influence the sensory processing of another. However, sensory processing is inherently adapted to encode a range of stimulus intensity. Thus, to characterize the enhancement of sensory integration in this study, varying stimulus intensity could be examined. Is the enhancement of sensory integration important for stimuli with weaker parameters than the ones used in the study? For stronger stimuli? Is the enhancement uniform across stimulus strength? This is a limitation worth addressing by mapping a contrast response function.

Related to this point, the decreased lick latency provided some evidence that the mouse is leveraging the auditory inputs as an aid in the task (rather than a distractor, as it is not needed for this task). Nonetheless, one would think that the auditory input would be more relevant if the tactile stimulus is presented at a level at or just above the detection threshold. Here it seems that the tactile stimulus is obvious to the mouse, therefore the performance is already at ceiling and multisensory integration is not particularly important.

- In the manuscript, the authors are modest about the novelty and significance of the findings. However, providing more context can be helpful for the readers. While the authors highlight that the critical function of the brain is to generally integrate information from multiple senses, the authors could further explain the significance that the enhancement and integration is observed at the single neuron level and among early sensory areas.

Minor comments:

- Adding line numbers would greatly facilitate the peer review process.
- Figure 1: How were L2/3 pyramidal neurons selectively and sparsely transfected? Purely by the depth of the virus injection – how do the authors ascertain that these are indeed L2/3 pyramidal neurons?
- Typo, p.3, reference {Oh, 2014 #49}
- The authors performed EMG recordings from neck muscles. However, it is still possible that movement occur elsewhere in the body. The mouse could be moving elsewhere in response to the auditory stimulus, and therefore is it possible dendritic signals arise from motor inputs. This is not to suggest further experiments, but note that it is a limitation and a caveat that should be noted in the Discussion.
- Figure 2E: a scatter plot may provide a better visualization for this data
- Figure 2E-F: Why is the enhanced somatic output during Aud+Tactile trials (Fig 2E) not reflected in somatic membrane potential (Fig 2F)? What is the interpretation?
- Figure 5H – should there be also auditory-tactile, unrewarded trials?
- The authors chose to use ArchT for photoinhibition of axonal inputs. There are unintended consequences to this approach, as Mahn et al. noted in their Nat Neurosci 2016 paper. The caveat and whether this would influence the author's interpretation of the data should be discussed.

Reviewer #2 (Remarks to the Author):

In the manuscript “Auditory input enhances somatosensory encoding and tactile goal-directed behavior”, the authors examined dendritic and somatic activity of forepaw S1 pyramidal neurons in response to auditory stimulus and activation of axons from auditory cortex. The authors found that auditory cortex input, when combined with somatosensory stimulation, produced enhanced activation of the L2/3 pyramidal neurons but not for the L5 pyramidal neurons, and the auditory sensory stimulus reduced the response time during a go/no-go tactile detection task. While I find that this appears to be a rigorously conducted study on an important problem of cellular mechanism for cross-modal integration, providing calcium imaging, intracellular recording and circuit manipulation evidence obtained from awake or behaving conditions, I have a few important concerns that I believe the authors would be able to address in a revised manuscript.

Major points:

1. The authors stated that for L2/3 neurons in S1, auditory stimuli only evoked dendritic responses without influencing somatic activity. This is a strong and surprising statement that would require more compelling evidence, for instance, by showing dendritic activity and somatic activity from the same neuron under the same sensory stimulation conditions. It has been shown that even in L5 neurons, distal dendritic activity under awake behaving conditions was almost always coupled with somatic activation. The L2/3 dendrites are much less compartmentalized comparing to L5 neurons, it would be very surprising that the large calcium signals the authors found in the dendrites (Fig. 1) were completely isolated from soma. For L2/3 neurons, the somatic activity can be easily imaged, especially under the sparse labeling condition the authors used, where both distal dendrites and soma from the same neuron can be imaged. However, the authors chose not to do so but to take greater pains to examine somatic responses using in vivo whole-cell recording in separate experiments (Fig.

2). This can bring significant confounds, e.g., given the small sample of somatic whole-cell recordings, it was possible that the neurons the authors recorded from were those that did not show any auditory evoked responses even in the dendrites, whereas the two-photon imaging experiments showing dendritic activation by auditory stimuli cannot exclude simultaneous somatic activation. Therefore, without imaging calcium signals from both distal dendrites and soma of the same L2/3 neurons, I do not think the authors would be able to conclude that the feedback input from auditory cortex only evokes local distal dendritic activity without influencing somatic output. ^[1]_[SEP]

2. The authors showed with the slice recording experiments that S1 L2/3 neurons exhibited subthreshold responses following stimulating the A1 axons (Fig. 3). But for in vivo whole-cell recordings (Fig. 2), why didn't the authors find any subthreshold activation following auditory stimulation? Assuming that there were dendritic activation by auditory stimuli, shouldn't such activation be manifest in subthreshold activity in somatic whole-cell recordings, especially in L2/3 neurons? In fact, this is where in vivo whole-cell recording could really help. But the authors only counted spike rate from the whole-cell recording data without quantifying subthreshold membrane potential changes. Counting spike rate would have been more easily obtained using extracellular recordings such as loose patch, or calcium imaging. ^[1]_[SEP]

3. The authors showed that the auditory cortex feedback input only influenced the dendrites of L2/3 neurons but not L5 neurons. It is rather surprising that the cortico-cortical feedback input in layer 1 has the specificity of only innervating L2/3 distal dendrites but avoiding the dendrites of L5 neurons. If this is indeed the case, the authors should cite prior studies with related evidence supporting such specificity in cortico-cortical connectivity. Alternatively, the authors should test or discuss other possibilities. For example, could there be stronger dendritic inhibition in the distal dendrites of L5 neurons (e.g., by SST neurons) that prevented dendritic activation by auditory cortex input? Or could it be that in naive animals, the auditory cortex input was not associated with significant behavioral events such that its impact on L5 neurons was negligible? Would auditory cortex input have any effect on L5 dendrites after behavioral conditioning? ^[1]_[SEP]

4. It is nice that the authors tested the behavioral effect of auditory cortex input to forepaw S1 using a tactile detection task. But there are caveats and limitations in the behavioral task design. The authors repeatedly claimed that this task was a goal-directed behavior (even in the title). But I am not convinced that this task can be justified as 'goal-directed' behavior, since it simply established a strong association between a single tactile stimulus and licking, which does not distinguish habitual vs. goal-directed mechanisms. Another limitation of this task is that it does not engage a mechanism to motivate multi-sensory integration, since the only rewarded trials were those contained tactile stimulus, while the presence of auditory stimulus was irrelevant to the behavioral outcomes. In fact, there was almost no behavioral effects following the addition of auditory stimulus other than a marginal decrease (from ~460 ms to ~433 ms) of licking latency. It is unclear what this small reduction in response time means in terms of sensory processing or sensorimotor transformation. One way to determine whether such decreased response time might be associated with reduced perceptual detection threshold is to vary the tactile stimulus intensity, and examine whether the addition of auditory stimulus would reduce the detection failure for weak tactile stimuli. ^[1]_[SEP]

5. The dendritic responses to auditory stimuli during task (Fig. 5) seems to be very sparse and unreliable, which is in contrast to those during non-task condition (Fig. 1). The authors may want to perform a better quantification of the calcium signals under auditory stimulus only and compare it with non-task condition.

6. Also in Fig. 5, only dendritic activity was shown for different types of stimuli. Were the response properties to different stimuli (auditory only, tactile only, auditory-tactile paired) specific to dendrites? If the authors were to conclude that this observation reflected a property of dendritic integration, it is important to also show the somatic activity during these stimulation conditions. Since for L2/3 neurons, soma can be readily imaged in the same field of view, this should not be difficult, and the authors might already have the data.

Minor points:

1. For imaging dendritic activity of L2/3 neurons (Figure 1), how did the author make sure the imaged dendrites belong to L2/3 neurons but not L5 neurons? Note that the virus could transduce both L2/3 and L5 neurons regardless of the injection depth. Given that the authors sparsely labeled these neurons, did the author verify somatic origin of the imaged dendrites by tracing down to the somatic region? If so, why didn't the authors provide somatic activity during auditory stimulation (see Major point 1)? ^[1]_{SEP}

2. It was nice that the authors performed in vivo pharmacology to block NMDAR using APV, and showed that this manipulation disrupted dendritic calcium signals. But this is not sufficient to support NMDA spikes, since there are many ways that NMDA receptor could contribute to dendritic depolarization and hence calcium signals other than generating NMDA spikes. ^[1]_{SEP}

3. Fig. 5 shows that tactile stimulus in hit trials with licking and reward produced much greater responses than the same tactile stimulus in miss trials, implying a contribution by reward or motor related feedback to the dendritic calcium signals. Did the authors examine dendritic activity following paired auditory and tactile stimulus in miss trials (non-rewarded) and compare it with auditory or tactile stimulation only condition in miss trials? ^[1]_{SEP}

4. Since both L5 and L2/3 neurons could be transfected by the GCaMP virus, did the author also image L5 neuron dendrites during behavior? How would L5 neuron dendritic activity be different from L2/3 neurons during the tactile detection task? Since large calcium spikes were previously observed for L5 distal dendrites during sensory detection task, it is likely that L5 dendritic activity could be more correlated with behavioral output. Unlike in naive animal, the multi-sensory integration that facilitates perceptual detection may contribute to L5 neuron dendritic spikes.

Reviewer #3 (Remarks to the Author):

Godenzini, Alwis et al. report on multi-sensory integration in somatosensory cortex of the mouse brain. Specifically, they find auditory responses in L2/3 pyramidal neurons of the forelimb region and enhanced tactile responses upon simultaneous auditory stimulation. Auditory stimulation alone was able to evoke dendritic calcium signals but did not increase the rate of somatic action potentials. Using optogenetic circuit mapping they identified direct inputs from auditory cortex onto layer 2/3 neurons in the forelimb area. Markedly, no such connections were found for layer 5 neurons and, consistently, these infragranular neurons did not show auditory modulation of dendritic signals and somatic action potential rates. Finally, the authors examined the relevance of this cross-modal integration by training mice in a mixed forepaw-auditory go-nogo task. They found that paired auditory stimulation enhanced dendritic response in forelimb L2/3 neurons in rewarded trials and that mice responded with a significantly shorter latency time when additional auditory stimulation was applied. Overall, the authors propose L2/3 distal dendrites as key integration site for combining information from separate sensory modalities.

I find this a highly original and relevant study that convincingly demonstrates auditory-tactile sensory integration in L2/3 dendrites. The experimental methods are state of the art, including imaging, in vitro and in vivo electrophysiology, optogenetics, and behavior. Data are of highest quality and the performed analyses are fully adequate. The manuscript is well written with clearly laid out figures. The demonstration of a behavioral effect is particularly impressive. The study is at the forefront of in vivo dendritic physiology and suggests direct cross-modal projections (at least in upper layers) as mechanism for sensory integration. I expect it to be of broad interest to neurophysiologists. Despite my overall positive assessment, I still have a number of questions and minor points that the authors should address.

Specific points:

1. What order of tuft dendrites were imaged? I presume imaging planes were above the main bifurcation but it would be helpful to obtain specific information whether mainly first-order or higher-

order branches were imaged.

2. How come that dendritic calcium signals in Figure 1 appear rather symmetric with slow onsets of around 0.5 s? Would one not expect sharp rises followed by slower decays?

3. How do auditory-evoked, respectively touch-evoked dendritic calcium signals in Fig. 1B (and AP output in Fig. 2b-d) depend on stimulus strength? Is the loudness maximal or could increasing the loudness possibly lift the L2/3 neurons above threshold for AP generation? Even though it is difficult to compare different modalities, how do the touch and tone stimuli employed compare in terms of saliency?

4. I miss reference to a previous study on auditory-evoked responses in rat barrel cortex: Maruyama and Komai, PLOS ONE 2018 (doi.org/10.1371/journal.pone.0209266).

5. Fig. 3a. What does the wider distribution of A1 axonal projections look like? Do they preferentially target FL and HL areas or the entire somatosensory map? From the reference mentioned one would also expect dense superficial axons in barrel cortex. Is further anatomical information on this aspect available, e.g., in the Allen Institute projection atlas?

6. Related to this point: How precise was targeting of the forepaw region and how were its boundaries demarcated? Figure S1 looks very rough in Fig S1. Was any functional validation attempted (intrinsic imaging or HL vs. FL stimulation)?

7. I find the marked difference between L2/3 and L5 neurons intriguing because they indicate that axons from auditory cortex wire up specifically with L2/3 dendrites. Is there any further evidence for such specificity? I was surprised that this difference is hardly treated in the Discussion. Perhaps the authors can expand on the discussion and put this finding into a broader context of L2/3-to-L5 difference in modulation, e.g. during locomotion (Ayaz et al., Nature Comm. 2019, doi.org/10.1038/s41467-019-10564-8).

8. Label dendrites in Fig. 5b explicitly as L2/3 dendrites to avoid confusion.

9. Page 7. It is unclear to me why a 200-ms earlier auditory tone onset was used as control and not a 200-ms later tone. 200 ms is not that long and given a 500-ms stimulation duration, the tone still overlapped with the tactile stimulus, correct? Moreover, the integration window might be longer than 200 ms (e.g. prolonged dendritic potentials) so that one might still expect a modulatory enhancing effect of auditory inputs, especially as it will be predictive of the touch. Thus, I did not expect a 200-ms earlier tone to not have an effect. Did the authors try other time delays?

We thank the reviewers for their assessment of our study and we have now performed new experiments and analysis which further support our findings and provides new insights into the multisensory integrative properties of cortical pyramidal neurons.

Reviewer #1 (Remarks to the Author):

Godenzini and colleagues investigated the cortical basis supporting the integration of multisensory information in the somatosensory cortex in mice.

The results add new insights to the body of work on multisensory integration. A striking finding is the selectivity of auditory inputs onto layer 2/3 but not layer 5 pyramidal neurons in the forepaw cortex. This distinction was demonstrated forcibly using in vitro and in vivo method. However, the authors did not quite fully explore the difference to show if the distinction is also behaviorally significant.

Notwithstanding this critique (which may be difficult to do for technical reasons), the study is solid. The experiments were performed carefully using an array of modern approaches. The results generate new understanding for how auditory inputs may be integrated in the somatosensory cortex of the mouse. The manuscript is well-written and logically organized.

We thank the reviewer for their assessment of our study.

Major comments:

- Figure 5 shows difference in dendritic calcium signals between Tactile-trials and AudTac-trials. While the authors intuit that this difference stems from the sensory component, it is possible that there may be contributions to these calcium responses from motor (licking) and/or outcome (sucrose-water reward) components. It is possible that the mouse may lick earlier and more vigorously during AudTac-trials, which would produce a motor-related difference. Additionally, a prior study showed substantial dendritic calcium responses in layer 2/3 pyramidal neurons in the barrel cortex in response to rewards (Lacefield et al., Cell Rep, 2019). For example, a contribution from reward may have an additive effect with combined auditory-tactile stimuli. If the authors can isolate the sensory-evoked calcium response from the reward component, they may see a greater enhancement for paired stimuli versus tactile stimuli. Can the contributions of sensory, motor, and outcome be dissociated? A linear regression analysis (e.g., Siniscalchi MJ et al., Cereb Cortex, 2019), with tactile stimulus, auditory stimulus, lick rate, and reward as factors, could be informative.

In the tactile goal-directed behavior tested in this study, both the Tactile-trials and AudTac-trials were rewarded with water reward. This was by design to enable the direct comparison of neural activity in response to different stimulus without the confound of reward delivery. Therefore, although reward delivery may contribute to the overall recorded signal, the differences in dendritic activity reported in this study were not due to water reward, since both trials types had the same reward delivery (HIT, 10 μ l). Since we were particularly interested in sensory integration and not reward signals, as a further precaution, we report only on dendritic events that occurred within 1 second from stimulus onset which excluded the reward time.

To address whether the difference in calcium responses were from motor (licking), we have now assessed whether the licking response to the Tactile-trials and AudTac-trials was different. This analysis indicated that the licking response was similar during Tactile-trials and AudTac-trials, indicating that the different dendritic activity during Tactile-trials and AudTac-trials was not due to difference in motor (licking) output. The following text is now included in the results: “This increase in tuft dendritic activity was not due to differences in licking behavior between the rewarded trials,

as there was no significant difference between the licking frequency during Tactile-trials and Audtac-trials (3.1 ± 0.4 Hz vs 3.1 ± 0.4 Hz; $n = 16$ mice, $p = 0.98$; Wilcoxon matched-pairs signed rank test).”

- Throughout the study, only one type of both auditory and tactile stimulus was used to show that the activation of one sense can influence the sensory processing of another. However, sensory processing is inherently adapted to encode a range of stimulus intensity. Thus, to characterize the enhancement of sensory integration in this study, varying stimulus intensity could be examined. Is the enhancement of sensory integration important for stimuli with weaker parameters than the ones used in the study? For stronger stimuli? Is the enhancement uniform across stimulus strength? This is a limitation worth addressing by mapping a contrast response function.

We agree with the reviewer that assessing the influence of auditory input over a range of stimulus intensities is of interest. We have now performed additional experiments where we have assessed the influence of auditory input on dendritic encoding using a range of stimuli. Since dendritic activity is sparse (evoked rate ~ 0.1), and we are limited by the number of trials that can be performed in a single session, we were unable to map a complete contrast response function.

1) We altered the intensity of the broadband auditory input over a range (50, 70, 90 dB) and recorded the dendritic response to auditory stimulus alone (Aud-stim) and combined with the tactile stimulus used in this study (500 ms, 200 Hz; AudTac-stim). As seen in new Supplementary Figure 3a, altering the intensity of the auditory stimulus did not influence the amplitude of the dendritic Ca^{2+} response evoked during Aud-stim and AudTac-stim. However, the auditory stimulus that had the lowest intensity (and was closest to the background noise) had the greatest influence on the rate of dendritic activity. We now include this information in the manuscript.

2) We altered the duration of the tactile stimulus over a range (75, 500, 1000 ms) and recorded the dendritic response to tactile stimulus alone (Tactile-stim) and combined with the auditory broadband stimulus used in the study (AudTac-stim). As seen in new Supplementary Figure 3b, altering the duration of the tactile stimulation did not influence the amplitude of the dendritic Ca^{2+} response evoked during Aud-stim and AudTac-stim. Although, the tactile stimulus with the shortest duration evoked significantly less dendritic Ca^{2+} transients, there was no significant influence of altering tactile stimulus on the evoked rate or amplitude of Ca^{2+} transients evoked during AudTac-stim. We now include this information in the manuscript.

Related to this point, the decreased lick latency provided some evidence that the mouse is leveraging the auditory inputs as an aid in the task (rather than a distractor, as it is not needed for this task). Nonetheless, one would think that the auditory input would be more relevant if the tactile stimulus is presented at a level at or just above the detection threshold. Here it seems that the tactile stimulus is obvious to the mouse, therefore the performance is already at ceiling and multisensory integration is not particularly important.

We agree with the reviewer that the performance may already be at maximal levels, which may reduce the influence of multisensory integration on the recorded behavior. We now discuss the possibility that multisensory stimuli may be more effective in altering behavior during less salient stimuli. Nevertheless, auditory input does alter sensory-based behavior during salient stimuli, suggesting an influence of multisensory integration even during obvious stimuli.

- In the manuscript, the authors are modest about the novelty and significance of the findings. However, providing more context can be helpful for the readers. While the authors highlight that the critical function of the brain is to generally integrate information from multiple senses, the authors could further explain the significance that the enhancement and integration is observed at the single neuron level and among early sensory areas.

We have now added additional information in the manuscript regarding the role of single neurons and the primary somatosensory cortex in integrating information from multiple senses.

“Integrating information from multiple senses is crucial to survival. The cortex is central to this process, connecting information from different senses in a layer and neuron specific manner. Here, we demonstrate that this can occur at the level of single neurons in the primary somatosensory cortex, influencing the sensory encoding in tuft dendrites of pyramidal neurons. This localized cellular enhancement of neural activity endows the cortex with multiple independent integrative units which may facilitate the coordination of activity across different cortical areas during sensory perception (Stein and Stanford, 2008) and act to increase cortical computational power (silver, 2016).”

Minor comments:

- Adding line numbers would greatly facilitate the peer review process.

This was an oversight and now have added line numbers to the manuscript.

- Figure 1: How were L2/3 pyramidal neurons selectively and sparsely transfected? Purely by the depth of the virus injection – how do the authors ascertain that these are indeed L2/3 pyramidal neurons?

L2/3 pyramidal neurons were sparsely transfected using a mix of Cre-dependent genetic Ca^{2+} indicator GCaMP6f (AAV1.Syn.Flex.GCaMP6f.WPRE.SV40) and diluted Cre (1:6000; AAV1.hSyn.Cre.WPRE.hGH) injected into layer 2/3 of the primary somatosensory cortex (at a depth of 450 μ m). In each experiment, we ascertain that L2/3 pyramidal neurons were indeed targeted by visualizing the somata of layer 2/3 pyramidal neurons within the injection site. This is now mentioned in the methods.

- Typo, p.3, reference {Oh, 2014 #49}

Corrected

- The authors performed EMG recordings from neck muscles. However, it is still possible that movement occur elsewhere in the body. The mouse could be moving elsewhere in response to the auditory stimulus, and therefore is it possible dendritic signals arise from motor inputs. This is not to suggest further experiments, but note that it is a limitation and a caveat that should be noted in the Discussion.

Since the mice are headfixed, large body movements would be reflected in the EMG. However, it is indeed possible that more subtle body movements are not measured in the neck muscles. This is now mentioned in the text.

- *Figure 2E: a scatter plot may provide a better visualization for this data*

In 2E, we illustrate the influence of auditory input on the tactile-evoked response in all recorded neurons. Due to variability in the number of evoked action potentials in the awake preparation use of a scatter plot would make it difficult to represent the data.

- *Figure 2E-F: Why is the enhanced somatic output during Aud+Tactile trials (Fig 2E) not reflected in somatic membrane potential (Fig 2F)? What is the interpretation?*

This is explained in detail in the discussion.

“Despite auditory input causing a significant increase in the tactile-evoked firing in L2/3 pyramidal neurons, there was no measurable influence of auditory input on the subthreshold response at the soma *in vivo*. This disconnect between subthreshold responses and enhanced action potential output has been reported previously in hindpaw S1 (Palmer, 2012) and is likely to be due to the generation of dendritic spikes in the distal apical dendrites, which can have a direct impact on action potential generation (Stuart, 1997; Larkum, 1999; Larkum, 2009; Palmer, 2014). Consistent with this idea, auditory input increased tactile-evoked Ca^{2+} transients in distal tuft dendrites of L2/3 pyramidal neurons.”

- *Figure 5H – should there be also auditory-tactile, unrewarded trials?*

During the tactile goal-directed behavior, mice performed at high rates of correct performance and did not often ‘miss’ responding to the AucTac stimulus. Therefore, these low rates of unrewarded AudTac trials did not allow for assessment of dendritic activity during incorrect behavior.

- *The authors chose to use ArchT for photoinhibition of axonal inputs. There are unintended consequences to this approach, as Mahn et al. noted in their Nat Neurosci 2016 paper. The caveat and whether this would influence the author’s interpretation of the data should be discussed.*

Since its development (Chow et al, 2009), the inhibitory opsin Archaeorhodopsin has been used to inhibit neurons from multiple brain regions (Dalmay et al, 2019; Quiquempoix et al, 2018). Although Mahn et al, 2016 reports ArchT can be excitatory, their protocol is considerably longer than the photoinhibition pulse in our study (5.5 sec vs 5 min). We now include text on the effectiveness of photoinhibition in the discussion.

“While long-duration (second) activation of ArchT can have excitatory effects (Mahn et al, 2016), activation of ArchT for shorter durations, as used in our study, has previously been shown to be inhibitory (Dalmay et al, 2019).”

Reviewer #2 (Remarks to the Author):

In the manuscript “Auditory input enhances somatosensory encoding and tactile goal-directed behavior”, the authors examined dendritic and somatic activity of forepaw S1 pyramidal neurons in response to auditory stimulus and activation of axons from auditory cortex. The authors found that auditory cortex input, when combined with somatosensory stimulation, produced enhanced activation of the L2/3 pyramidal neurons but not for the L5 pyramidal neurons, and the auditory sensory stimulus reduced the response time during a go/no-go tactile detection task. While I find that this appears to be a rigorously conducted study on an important problem of cellular mechanism for cross-modal integration, providing calcium imaging, intracellular recording and circuit manipulation evidence obtained from awake or behaving conditions, I have a few important concerns that I believe the authors would be able to address in a revised manuscript.

We thank the reviewer for their assessment of the manuscript.

Major points:

1. The authors stated that for L2/3 neurons in S1, auditory stimuli only evoked dendritic responses without influencing somatic activity. This is a strong and surprising statement that would require more compelling evidence, for instance, by showing dendritic activity and somatic activity from the same neuron under the same sensory stimulation conditions. It has been shown that even in L5 neurons, distal dendritic activity under awake behaving conditions was almost always coupled with somatic activation. The L2/3 dendrites are much less compartmentalized comparing to L5 neurons, it would be very surprising that the large calcium signals the authors found in the dendrites (Fig. 1) were completely isolated from soma. For L2/3 neurons, the somatic activity can be easily imaged, especially under the sparse labeling condition the authors used, where both distal dendrites and soma from the same neuron can be imaged. However, the authors chose not to do so but to take greater pains to examine somatic responses using in vivo whole-cell recording in separate experiments (Fig. 2). This can bring significant confounds, e.g., given the small sample of somatic whole-cell recordings, it was possible that the neurons the authors recorded from were those that did not show any auditory evoked responses even in the dendrites, whereas the two-photon imaging experiments showing dendritic activation by auditory stimuli cannot exclude simultaneous somatic activation. Therefore, without imaging calcium signals from both distal dendrites and soma of the same L2/3 neurons, I do not think the authors would be able to conclude that the feedback input from auditory cortex only evokes local distal dendritic activity without influencing somatic output.

Since cortical pyramidal neurons sparsely encode sensory information and typically fire less than one action potential in response to tactile stimulus, we performed whole-cell patch clamp recordings in this study as this technique has high (action potential) resolution. Although the reviewer is correct in pointing out we were only able to record from a small number of neurons compared to the number of neurons within layer 2/3, however, in all neurons (n = 19), there was not a measurable influence of auditory stimulus at either the suprathreshold or subthreshold level. It is not possible to ascertain subthreshold voltage activity using *in vivo* Ca²⁺ imaging.

Unfortunately, the sparse labelling technique required for dendritic imaging does not alleviate the issue of small sample sizes as we typically only label a few neurons per injection. Therefore, we have now investigated auditory-evoked responses in layer 2/3 pyramidal neuron somata using dense labelling (AAV.GCaMP6f.WPRE.SV40). Similar to voltage recordings, auditory input alone did not reliably generate somatic Ca²⁺ transients. However, there was a significant increase in evoked activity when the auditory input was paired with a tactile stimulus. This new data is included as a new Supplementary Figure 6.

Comparing the rate of Ca²⁺ transients in the soma and tuft dendrites during auditory input in neurons densely labelled with the calcium indicator, GCaMP6f, illustrates that, within the same population of neurons, tuft dendrites had significantly greater auditory-evoked activity than layer 2/3 somata (new Supplementary Figure 6d). Combined with the patch clamp voltage recordings, this disconnect between dendritic and somatic auditory-evoked activity further suggest that local distal dendritic activity does not always result in somatic output.

2. The authors showed with the slice recording experiments that S1 L2/3 neurons exhibited subthreshold responses following stimulating the A1 axons (Fig. 3). But for in vivo whole-cell recordings (Fig. 2), why didn't the authors find any subthreshold activation following auditory stimulation? Assuming that there were dendritic activation by auditory stimuli, shouldn't such activation be manifest in subthreshold activity in somatic whole-cell recordings, especially in L2/3 neurons? In fact, this is where in vivo whole-cell recording could really help. But the authors only counted spike rate from the whole-cell recording data without quantifying subthreshold membrane potential changes. Counting spike rate would have been more easily obtained using extracellular recordings such as loose patch, or calcium imaging.

We agree, counting spike rate would have been easier to obtain using extracellular recordings, however, we were interested in also recording the evoked subthreshold activity as this can tell us a lot about the overall synaptic input (see Palmer et al, 2014).

During the *in vivo* patch clamp recordings, on average there was no measurable subthreshold voltage response. This is in contrast to the *in vitro* recordings from layer 2/3 pyramidal neurons, which report subthreshold responses to photoactivation of auditory input. The discrepancy between the *in vitro* and *in vivo* subthreshold results could be due to the photoactivation of axons from the auditory cortex recruiting many more inputs than auditory stimuli and does so in a highly synchronized way (compared to the auditory input which might have more jitter). This is now discussed in the discussion.

3. The authors showed that the auditory cortex feedback input only influenced the dendrites of L2/3 neurons but not L5 neurons. It is rather surprising that the cortico-cortical feedback input in layer 1 has the specificity of only innervating L2/3 distal dendrites but avoiding the dendrites of L5 neurons. If this is indeed the case, the authors should cite prior studies with related evidence supporting such specificity in cortico-cortical connectivity. Alternatively, the authors should test or discuss other possibilities. For example, could there be stronger dendritic inhibition in the distal dendrites of L5 neurons (e.g., by SST neurons) that prevented dendritic activation by auditory cortex input? Or could it be that in naive animals, the auditory cortex input was not associated with significant behavioral events such that its impact on L5 neurons was negligible? Would auditory cortex input have any effect on L5 dendrites after behavioral conditioning?

We did not observe a significant decrease in evoked activity compared to spontaneous levels in L5 neurons during auditory input. This suggests that auditory input did not lead to significant dendritic inhibition in L5 neurons. With regard to the Reviewer's second point, in L2/3 pyramidal neurons auditory input influenced sensory processing in both naïve (Figure 1) and expert (Figure 5) mice. Therefore, we predict there would be no change in auditory input efficacy in L5 pyramidal neurons following behavioral conditioning.

4. It is nice that the authors tested the behavioral effect of auditory cortex input to forepaw S1 using a tactile detection task. But there are caveats and limitations in the behavioral task design. The

authors repeatedly claimed that this task was a goal-directed behavior (even in the title). But I am not convinced that this task can be justified as 'goal-directed' behavior, since it simply established a strong association between a single tactile stimulus and licking, which does not distinguish habitual vs. goal-directed mechanisms. Another limitation of this task is that it does not engage a mechanism to motivate multi-sensory integration, since the only rewarded trials were those contained tactile stimulus, while the presence of auditory stimulus was irrelevant to the behavioral outcomes. In fact, there was almost no behavioral effects following the addition of auditory stimulus other than a marginal decrease (from ~460 ms to ~433 ms) of licking latency. It is unclear what this small reduction in response time means in terms of sensory processing or sensorimotor transformation. One way to determine whether such decreased response time might be associated with reduced perceptual detection threshold is to vary the tactile stimulus intensity, and examine whether the addition of auditory stimulus would reduce the detection failure for weak tactile stimuli.

We were careful in our use of terminology and ensured that our behavior was indeed goal-directed and not habitual. To illustrate this, we performed extinction experiments (no water reward) and mice were able to extinguish the tactile Go/NoGo behavior rapidly within a single session. Therefore, we conclude that this behavior is goal-directed and we now mention this in the Methods.

The behavior used in this study was designed to purely investigate the influence of auditory input on the processing of tactile information in forepaw S1. By rewarding both Tactile and AudTac trials, we assessed the influence of auditory input on processing of tactile sensory information while avoiding the confound of water reward delivery. We agree that the behavioral effect is small, however, taking into account the reaction time (200 ms; Wong et al, 2015), the 30 ms difference equates to a substantial portion of the response time (~ 15 %). We now include more detail in the discussion regarding this decreased response time.

We have now performed additional experiments where we decreased the duration of the tactile stimulus to the average perception threshold measured in our system (75 ms; 200 Hz). Decreasing the tactile stimulus duration to this perceptual-threshold resulted in a decrease in the evoked rate of dendritic Ca²⁺ transients, although there was no difference in the influence of auditory input during the perceptual-threshold tactile stimulus and that used in this study (500 ms). This data is included in a new Supplementary Figure 3. While there was no detectable influence of auditory input on dendritic activity evoked by tactile stimuli of different durations, it is possible that auditory input would influence the perception of near-threshold stimuli during behavior when the brain is more active.

5. The dendritic responses to auditory stimuli during task (Fig. 5) seems to be very sparse and unreliable, which is in contrast to those during non-task condition (Fig. 1). The authors may want to perform a better quantification of the calcium signals under auditory stimulus only and compare it with non-task condition.

Dendrites are typically sparsely active. The probability of auditory stimuli evoking a dendritic response was not significantly different during the task (0.10 ± 0.001 , $n = 70$ dendrites) and non-task (0.09 ± 0.005 , $n = 110$ dendrites) conditions ($p = 0.476$, Mann Whitney Test).

6. Also in Fig. 5, only dendritic activity was shown for different types of stimuli. Were the response properties to different stimuli (auditory only, tactile only, auditory-tactile paired) specific to dendrites? If the authors were to conclude that this observation reflected a property of dendritic integration, it is important to also show the somatic activity during these stimulation conditions. Since for L2/3 neurons, soma can be readily imaged in the same field of view, this should not be difficult, and the authors might already have the data.

We have now compared the auditory-evoked responses in the dendrites and somata of layer 2/3 pyramidal neurons. In contrast to dendritic responses, 1) there was no significant difference in evoked activity between baseline and auditory stimulus at the soma, 2) there was a significant increase in evoked activity when the auditory input was paired with tactile stimulus, 3) there was considerably less somatic activity during the auditory stimulus alone compared to when it was paired with the tactile stimulus and 4) there was greater evoked activity in tuft dendrites compared with soma. This data is now included as a new Supplementary Figure 6.

Minor points:

1. For imaging dendritic activity of L2/3 neurons (Figure 1), how did the author make sure the imaged dendrites belong to L2/3 neurons but not L5 neurons? Note that the virus could transduce both L2/3 and L5 neurons regardless of the injection depth. Given that the authors sparsely labeled these neurons, did the author verify somatic origin of the imaged dendrites by tracing down to the somatic region? If so, why didn't the authors provide somatic activity during auditory stimulation (see Major point 1)?

In each experiment, we ascertain that L2/3 pyramidal neurons were indeed targeted by visualizing the entire dendritic tree and the somata of the recorded neurons within the injection site. The following is now included in the Methods:

“Layer specificity of the injection site and identification of the recorded cell-type was confirmed after each experiment by visualizing the entire dendritic arbour and location of the associated somata.”

2. It was nice that the authors performed in vivo pharmacology to block NMDAr using APV, and showed that this manipulation disrupted dendritic calcium signals. But this is not sufficient to support NMDA spikes, since there are many ways that NMDA receptor could contribute to dendritic depolarization and hence calcium signals other than generating NMDA spikes.

We acknowledge that it is extremely difficult to definitely categorize calcium signals as NMDA spikes (see Palmer et al., 2014). We have therefore removed the suggestion of NMDA spikes from the manuscript.

3. Fig. 5 shows that tactile stimulus in hit trials with licking and reward produced much greater responses than the same tactile stimulus in miss trials, implying a contribution by reward or motor related feedback to the dendritic calcium signals. Did the authors examine dendritic activity following paired auditory and tactile stimulus in miss trials (non-rewarded) and compare it with auditory or tactile stimulation only condition in miss trials?

We do not report miss trials throughout the manuscript as the number of miss trials was relatively small and not enough for analysis. Figure 5 illustrates that during correct trials (Hit), both the rate and amplitude of Ca^{2+} transients was greater during AudTac-trials compared with Tactile-trials. In the behavior tested, the tactile stimulus was rewarded with and without an auditory stimulus (ie, both Tactile-trials and Audtac-trials) if the mouse licked during the response window (1 sec). Only trials that had the auditory stimulus alone (Aud-trials) were not rewarded. Since both Tactile-trials and AudTac-trials involve the same behavior (ie licking in response to the stimulus to receive a water reward), any contribution of reward or motor-related feedback to the response would be the same on both these trial types. We conclude, therefore, that the increased dendritic activity during AudTac-trials is not due to reward delivery. We have now analyzed the licking responses during AudTac-trials and Tactile-trials, and find there was no significant difference between the licking frequency

during Tac-trials and AudTac-trials (3.1 ± 0.4 Hz vs 3.1 ± 0.4 Hz; $n = 16$ mice, $p = 0.98$; Wilcoxon matched-pairs signed rank test). This information is now included in the manuscript. This new analysis supports the idea that the increased dendritic activity during AudTac-trials does not result from a different behavioral output. We have included a new schematic in the revised manuscript which better illustrates the experimental behavioral paradigm (new Figure 5A).

4. Since both L5 and L2/3 neurons could be transfected by the GCaMP virus, did the author also image L5 neuron dendrites during behavior? How would L5 neuron dendritic activity be different from L2/3 neurons during the tactile detection task? Since large calcium spikes were previously observed for L5 distal dendrites during sensory detection task, it is likely that L5 dendritic activity could be more correlated with behavioral output. Unlike in naive animal, the multi-sensory integration that facilitates perceptual detection may contribute to L5 neuron dendritic spikes.

As L5 pyramidal neurons do not receive direct input from the auditory cortex we did not perform Ca^{2+} imaging from L5 dendrites during behavior as we did not feel adding this data would significantly advance the manuscript.

Reviewer #3 (Remarks to the Author):

Godenzini, Alwis et al. report on multi-sensory integration in somatosensory cortex of the mouse brain. Specifically, they find auditory responses in L2/3 pyramidal neurons of the forelimb region and enhanced tactile responses upon simultaneous auditory stimulation. Auditory stimulation alone was able to evoke dendritic calcium signals but did not increase the rate of somatic action potentials. Using optogenetic circuit mapping they identified direct inputs from auditory cortex onto layer 2/3 neurons in the forelimb area. Markedly, no such connections were found for layer 5 neurons and, consistently, these infragranular neurons did not show auditory modulation of dendritic signals and somatic action potential rates. Finally, the authors examined the relevance of this cross-modal integration by training mice in a mixed forepaw-auditory go-nogo task. They found that paired auditory stimulation enhanced dendritic response in forelimb L2/3 neurons in rewarded trials and that mice responded with a significantly shorter latency time when additional auditory stimulation was applied. Overall, the authors propose L2/3 distal dendrites as key integration site for combining information from separate sensory modalities.

I find this a highly original and relevant study that convincingly demonstrates auditory-tactile sensory integration in L2/3 dendrites. The experimental methods are state of the art, including imaging, in vitro and in vivo electrophysiology, optogenetics, and behavior. Data are of highest quality and the performed analyses are fully adequate. The manuscript is well written with clearly laid out figures. The demonstration of a behavioral effect is particularly impressive. The study is at the forefront of in vivo dendritic physiology and suggests direct cross-modal projections (at least in upper layers) as mechanism for sensory integration. I expect it to be of broad interest to neurophysiologists. Despite my overall positive assessment, I still have a number of questions and minor points that the authors should address.

We thank the reviewer for their assessment and have now altered the manuscript to address their questions and minor points.

Specific points:

1. *What order of tuft dendrites were imaged? I presume imaging planes were above the main bifurcation but it would be helpful to obtain specific information whether mainly first-order or higher-order branches were imaged.*

All imaging from tuft dendrites was performed in higher-order branches beyond the bifurcation point. This information is now included in the methods.

2. *How come that dendritic calcium signals in Figure 1 appear rather symmetric with slow onsets of around 0.5 s? Would one not expect sharp rises followed by slower decays?*

Similar symmetrical responses were recorded in cortical tuft dendrites *in vivo* in previous studies (Xu et al., 2012; Chen et al., 2013) which may be due to the high affinity of the calcium indicator, GCaMP, which results in the kinetics of the fluorescence not necessarily following the underlying voltage *in vivo*. Furthermore, as reported in the methods, Ca²⁺ responses were filtered using a Savitzky-Golay filter with a 2nd order polynomial which smoothed the signal.

3. *How do auditory-evoked, respectively touch-evoked dendritic calcium signals in Fig. 1B (and AP output in Fig. 2b-d) depend on stimulus strength? Is the loudness maximal or could increasing the loudness possibly lift the L2/3 neurons above threshold for AP generation? Even though it is difficult to compare different modalities, how do the touch and tone stimuli employed compare in terms of saliency?*

We have now performed experiments where we altered the intensity of the broadband auditory input (50, 70, 90 dB) and recorded the dendritic response to auditory stimulus alone (Aud-stim). As seen in new Supplementary Figure 3a, altering the intensity of the auditory stimulus did not influence the amplitude of the evoked dendritic Ca²⁺ response, and the auditory stimulus that had the lowest intensity (and was closest to the background noise) had the greatest influence on the rate of dendritic activity.

In terms of saliency, both touch and tone stimuli employed in this study resulted in similar pupil dilation (Supplementary Figure 5) and were above threshold for sensory perception, which is now stated in the methods.

4. *I miss reference to a previous study on auditory-evoked responses in rat barrel cortex: Maruyama and Komai, PLOS ONE 2018 (doi.org/10.1371/journal.pone.0209266).*

This was an oversight and the reference is now included in the manuscript.

5. *Fig. 3a. What does the wider distribution of A1 axonal projections look like? Do they preferentially target FL and HL areas or the entire somatosensory map? From the reference mentioned one would also expect dense superficial axons in barrel cortex. Is further anatomical information on this aspect available, e.g., in the Allen Institute projection atlas?*

According to the Allen Institute projection atlas, the auditory cortex projects to various subcortical and cortical regions including the thalamus, striatum, posterior parietal cortex, and visual cortex to

name a few. Although it is unclear whether they preferentially target specific areas, these various pathways may have different influences on multisensory integration (Iurilli et al, 2012).

6. Related to this point: How precise was targeting of the forepaw region and how were its boundaries demarcated? Figure S1 looks very rough in Fig S1. Was any functional validation attempted (intrinsic imaging or HL vs. FL stimulation)?

The forepaw region was targeted according to the coordinates obtained from the Allen Institute projection atlas. This location was confirmed using whole-cell patch clamp recordings with a forepaw response evoked in all cells within this region. For calcium imaging, since we used a sparse labelling technique, the boundaries of the recorded area were defined by / restricted to the injection site.

7. I find the marked difference between L2/3 and L5 neurons intriguing because they indicate that axons from auditory cortex wire up specifically with L2/3 dendrites. Is there any further evidence for such specificity? I was surprised that this difference is hardly treated in the Discussion. Perhaps the authors can expand on the discussion and put this finding into a broader context of L2/3-to-L5 difference in modulation, e.g. during locomotion (Ayaz et al., Nature Comm. 2019, doi.org/10.1038/s41467-019-10564-8).

We agree that these results are of great interest and we have now expanded the discussion on this issue and included the suggested publication.

8. Label dendrites in Fig. 5b explicitly as L2/3 dendrites to avoid confusion.

We have now labeled Figure 5b to avoid confusion.

9. Page 7. It is unclear to me why a 200-ms earlier auditory tone onset was used as control and not a 200-ms later tone. 200 ms is not that long and given a 500-ms stimulation duration, the tone still overlapped with the tactile stimulus, correct? Moreover, the integration window might be longer than 200 ms (e.g. prolonged dendritic potentials) so that one might still expect a modulatory enhancing effect of auditory inputs, especially as it will be predictive of the touch. Thus, I did not expect a 200-ms earlier tone to not have an effect. Did the authors try other time delays?

The auditory stimulus was presented 200 ms before the tactile stimulus to investigate whether auditory input evoked long-lasting GABA_B-mediated inhibition. Here, since GABA_B-mediated inhibition is typically seen hundreds of milliseconds after release, the auditory input needed to occur ~ 200 ms prior to the tactile stimulus to have a measurable influence on sensory encoding. However, since this is not expanded upon in the manuscript, for clarity, we have now removed this data from the manuscript.

REVIEWER COMMENTS

Reviewer #1 (Remarks to the Author):

I would emphasize again on the logical and carefully performed experiments, which provide new insights into how single neurons in the cortex integrate different streams of sensory inputs.

In response to the comments, the authors performed additional experiments, including addressing the topic of stimulus response function by testing several stimulus duration and intensity. I agree with the authors that this is not easy to map, because dendritic response is sparse, and applaud their effort to add these results.

They also provide additional analyses which argue against a potential motor confound that may account for differences in observed dendritic activity across trial type.

Overall, the reviewers addressed the comments. I remain convinced that this is an excellent study that should be published soon.

Reviewer #2 (Remarks to the Author):

In this revision, the authors have provided some new data and added additional discussions to clarify some of the issues raised by the reviewers. However, several of my previous concerns are still not addressed.

1. The authors still do not have solid evidence to support their claim that auditory cortex input to S1 only produces local dendritic activity restricted within the distal dendritic compartment in L2/3 S1 neurons. As I stated in my previous comments, point 1, "Without imaging calcium signals from both distal dendrites and soma of the same L2/3 neurons, I do not think the authors would be able to conclude that the feedback input from auditory cortex only evokes local distal dendritic activity without influencing somatic output." What the authors should do, in order to address this question, is that when they find dendritic activity evoked by auditory stimuli, they should then focus down to the soma of the same neuron, and determine whether the same auditory stimuli could also evoke somatic responses. Such experiment is feasible under the sparse labeling condition, since the authors mentioned in the Methods "Layer specificity of the injection site and identification of the recorded cell-type was confirmed after each experiment by visualizing the entire dendritic arbour and location of the associated somata."

The authors only provided comparison of calcium signals between dendrites and soma of densely labeled L2/3 neuron populations. There could be important confounds in such comparison. The dendrites and soma can be from completely different populations. The different event rates in these two populations cannot support local dendritic activity restricted in the distal dendritic compartment. Each neuron can have many dendritic branches in the upper layer, it is very different sampling when imaging from tuft branches and soma at population level, and directly comparing the event rates can be misleading. It is also unclear whether the distal dendrites they imaged were all from L2/3 neurons, since the somatic origin cannot be verified under dense labeling condition, while AAV injection could transfect both L2/3 and L5 neurons regardless of the injection depth.

2. The evidence to support the claim that L2/3 neurons integrate multi-sensory information during goal-directed behavior is still weak.

For the calcium imaging data, in Figure 1, there was a marginal enhancement of dendritic response under "AudTac" stimulation comparing to "Tac" only. However, the more crucial evidence for integration should be the enhancement of somatic activity under multi-sensory stimulation, which was not shown in the experiments in Figure 1. In the current revision, the authors provided some new somatic calcium imaging data in Supplementary Figure 6. But the authors have not explicitly

compared the somatic responses to “AudTac” stimuli vs “Tac” only stimuli, and used this to support the multi-sensory integration in L2/3 neurons.

For the voltage recording data shown in Figure 2, the significance in enhancement of spike rate under “AudTac” condition appears to be mainly due to two outlier neurons deviant from the main population (Fig 2e). Would there still be significant enhancement in the main population of sampled neurons without these two outliers? In the meanwhile, the evoked somatic voltage does not show significant enhancement under “AudTac” stimulation, which should be explained.

The L2/3 imaging data shown in Figure 4 appear to be contradictory with those shown in Figure 1. Apparently the calcium event rate is not significantly different between AudTac and Aud stimulation (Fig. 4h).

Under behaving condition, again the authors only provided data showing enhanced dendritic responses under AudTac vs Aud (Figure 5), but no data for somatic activity. To support the multi-sensory integration in L2/3 neurons during behavioral task, the somatic activity showing enhanced responses to multi-sensory stimulation during behavioral task is necessary.

Overall, the evidence for multi-sensory integration in S1 L2/3 neurons, especially under behaving condition is still weak in this revision. The authors need to either strengthen the evidence or revise their conclusions to reflect the limitations in their supporting data.

3. As I mentioned in my previous comments (point 3), in order to support the strong claim that “Auditory input does not influence dendritic integration and somatic output in L5 pyramidal neurons”, it is not enough to only examine L5 dendritic activity under passive stimulation. Dendritic integration can be highly dependent on behavioral state. It has been well-documented that cortical activity can be very different between passive stimulation and active behaving conditions. One cannot predict what would happen under behaving conditions simply based on results from passive stimulation. The authors should at least image dendritic calcium signals of L5 neurons under the same behavior conditions where they observed AudTac enhanced responses in L2/3 neurons. Otherwise, they should explicitly state that their conclusions about L5 neurons only apply to passive stimulation condition, but not task performance conditions.

4. In my previous comments, point 4, “One way to determine whether such decreased response time might be associated with reduced perceptual detection threshold is to vary the tactile stimulus intensity, and examine whether the addition of auditory stimulus would reduce the detection failure for weak tactile stimuli.”

The authors missed the key point in their response. The key point is to determine the behavior effect of concurrent auditory stimulation on threshold-level tactile detection. However, the authors only examined the effect of varying stimulus intensity on the activity of L2/3 neurons but not on behavioral performance. The more important question of whether multi-sensory input in the current behavioral task has any effect on the performance of tactile detection remains unaddressed.

5. In my previous comments (point 6), when I was asking about the somatic responses, I was referring to the data in Fig. 5, which is from behavioral task. But the new data of somatic activity shown in Supplementary Fig 6 are from passive stimulation. It is important to show that multi-sensory stimulation enhances somatic activity during a task where such multi-sensory information is relevant, as also mentioned above.

Reviewer #3 (Remarks to the Author):

The authors have addressed the various points of critique well and I find the revised manuscript much improved. They have answered most of my questions sufficiently but I still like to make two remarks:

Ad 3) I find it puzzling that, when varying the loudness of the auditory stimulation, the lowest sound intensity has the largest effect. The authors should at least comment on this peculiar finding and provide their interpretation.

Ad 5) The authors did not answer my question regarding the distribution across FL and HL areas and

the comparison with barrel cortex. Perhaps they still have not consulted the Allen atlas in detail, which they should do. They should pull out all the relevant Allen experiments (projections from A1 source to FL/HL/BC targets), look at the 3D histology datasets and convey to the reader what they find in terms of axonal projection patterns across areas and layers. It is also easy to extract from the Allen atlas the quantifications for these projections in terms of axonal fluorescence in the target regions.

Reviewer #1 (Remarks to the Author):

I would emphasize again on the logical and carefully performed experiments, which provide new insights into how single neurons in the cortex integrate different streams of sensory inputs.

In response to the comments, the authors performed additional experiments, including addressing the topic of stimulus response function by testing several stimulus duration and intensity. I agree with the authors that this is not easy to map, because dendritic response is sparse, and applaud their effort to add these results.

They also provide additional analyses which argue against a potential motor confound that may account for differences in observed dendritic activity across trial type.

Overall, the reviewers addressed the comments. I remain convinced that this is an excellent study that should be published soon.

We thank the reviewer for their comments and are enthused by their support.

Reviewer #2 (Remarks to the Author):

In this revision, the authors have provided some new data and added additional discussions to clarify some of the issues raised by the reviewers. However, several of my previous concerns are still not addressed.

We thank the reviewer for their comments and suggestions and have now performed additional experiments and analysis to address the concerns.

1. The authors still do not have solid evidence to support their claim that auditory cortex input to S1 only produces local dendritic activity restricted within the distal dendritic compartment in L2/3 S1 neurons. As I stated in my previous comments, point 1, "Without imaging calcium signals from both distal dendrites and soma of the same L2/3 neurons, I do not think the authors would be able to conclude that the feedback input from auditory cortex only evokes local distal dendritic activity without influencing somatic output." What the authors should do, in order to address this question, is that when they find dendritic activity evoked by auditory stimuli, they should then focus down to the soma of the same neuron, and determine whether the same auditory stimuli could also evoke somatic responses. Such experiment is feasible under the sparse labeling condition, since the authors mentioned in the Methods "Layer specificity of the injection site and identification of the recorded cell- type was confirmed after each experiment by visualizing the entire dendritic arbour and location of the associated somata."

The authors only provided comparison of calcium signals between dendrites and soma of densely labeled L2/3 neuron populations. There could be important confounds in such comparison. The dendrites and soma can be from completely different populations. The different event rates in these two populations cannot support local dendritic activity restricted in the distal dendritic compartment. Each neuron can have many dendritic branches in the upper layer, it is very different sampling when imaging from tuft branches and soma at population level, and directly comparing the event rates can be misleading. It is also unclear

whether the distal dendrites they imaged were all from L2/3 neurons, since the somatic origin cannot be verified under dense labeling condition, while AAV injection could transfect both L2/3 and L5 neurons regardless of the injection depth.

We have now performed experiments where we have, as suggested by the reviewer, measured both dendritic and somatic activity from the same sparsely labelled neurons. As shown in new Supplementary Figure 6, similarly to what we previously observed in densely labelled conditions, auditory-evoked Ca^{2+} responses in tuft dendrites of L2/3 pyramidal neurons were significantly greater in amplitude and probability than at the soma of the same cells (tuft, 1.04 ± 0.09 vs soma, 0.55 ± 0.2 , $p < 0.0001$, Mann Whitney test; $n = 116$ tuft dendrites, 38 soma; 6 mice). We also show an example of tuft dendritic and somatic activity in a single reconstructed neuron which clearly illustrates auditory-evoked activity in the tuft dendrites, and not at the soma (Supplementary Figure 6f and g). It is worth noting that, although feasible, such paired recordings are low yield using the sparse labelling technique.

2. The evidence to support the claim that L2/3 neurons integrate multi-sensory information during goal-directed behavior is still weak. For the calcium imaging data, in Figure 1, there was a marginal enhancement of dendritic response under “AudTac” stimulation comparing to “Tac” only. However, the more crucial evidence for integration should be the enhancement of somatic activity under multi-sensory stimulation, which was not shown in the experiments in Figure 1. In the current revision, the authors provided some new somatic calcium imaging data in Supplementary Figure 6. But the authors have not explicitly compared the somatic responses to “AudTac” stimuli vs “Tac” only stimuli, and used this to support the multi-sensory integration in L2/3 neurons.

In the revised manuscript, we now include a direct comparison between the somatic Ca^{2+} response to Tactile and AudTac stimulus (new panel d in Supplementary Figure 6). There was a significant increase in the rate of tactile-evoked somatic Ca^{2+} responses when paired with auditory input (Tactile, 0.06 ± 0.01 vs 0.08 ± 0.01 , $p = 0.03$ Wilcoxon matched-pairs signed rank test).

For the voltage recording data shown in Figure 2, the significance in enhancement of spike rate under “AudTac” condition appears to be mainly due to two outlier neurons deviant from the main population (Fig 2e). Would there still be significant enhancement in the main population of sampled neurons without these two outliers? In the meanwhile, the evoked somatic voltage does not show significant enhancement under “AudTac” stimulation, which should be explained

Action potential output can be extremely variable in the awake state. This variability is not obvious in the data shown in Fig. 2e as the data are normalized to the “Tac” alone condition. Nevertheless, removal of the two neurons with the highest increase in spiking rate during “AudTac” input still resulted in a significant increase in action potential output when the tactile and auditory stimulus were paired compared to the tactile stimulus alone ($p = 0.035$).

In lines 363 – 366 of the discussion, we explain the disconnect between subthreshold responses and enhanced action potential output. We now also include a statement in the results of the revised manuscript which mentions the importance of the results, where we say: “Therefore,

as reported previously³⁴, the increase in firing during paired auditory and tactile stimulus is not simply due to increased summation of synaptic input at the soma.”

The L2/3 imaging data shown in Figure 4 appear to be contradictory with those shown in Figure 1. Apparently the calcium event rate is not significantly different between AudTac and Aud stimulation (Fig. 4h).

We thank the reviewer for pointing out this error. We have now revised Figure 4h, and unlike Figure 1, here the data must be normalized to enable direct comparison between L2/3 and L5 pyramidal neurons. Previously, both L2/3 and L5 dendrites were normalized to the average tactile response per dendrite. However, since they are heterogeneously-responding populations (where some dendrites have low evoked activity), the data in Figure 4h is now normalised to the overall average tactile response for each dendritic region. In agreement with the data shown in Figure 1, there is a significant increase in the rate of normalized tactile-evoked responses during AudTac compared to Aud stimulus alone in L2/3 neurons ($p < 0.0001$, Kruskal-Wallis test). This is now described in the methods.

Under behaving condition, again the authors only provided data showing enhanced dendritic responses under AudTac vs Aud (Figure 5), but no data for somatic activity. To support the multi-sensory integration in L2/3 neurons during behavioral task, the somatic activity showing enhanced responses to multi-sensory stimulation during behavioral task is necessary.

Overall, the evidence for multi-sensory integration in S1 L2/3 neurons, especially under behaving condition is still weak in this revision. The authors need to either strengthen the evidence or revise their conclusions to reflect the limitations in their supporting data.

We now performed new experiments where we recorded the activity of L2/3 somata during behavior (new Supplementary Figure 11). Here, similar to tuft dendrites, somatic activity is enhanced during the AudTac-trials in the behavioral task.

3. As I mentioned in my previous comments (point 3), in order to support the strong claim that “Auditory input does not influence dendritic integration and somatic output in L5 pyramidal neurons”, it is not enough to only examine L5 dendritic activity under passive stimulation. Dendritic integration can be highly dependent on behavioral state. It has been well-documented that cortical activity can be very different between passive stimulation and active behaving conditions. One cannot predict what would happen under behaving conditions simply based on results from passive stimulation. The authors should at least image dendritic calcium signals of L5 neurons under the same behavior conditions where they observed AudTac enhanced responses in L2/3 neurons. Otherwise, they should explicitly state that their conclusions about L5 neurons only apply to passive stimulation condition, but not task performance conditions.

We agree with the reviewer and now include in the discussion that L5 pyramidal neuron tuft dendrites may act differently during behavior.

4. *In my previous comments, point 4, “One way to determine whether such decreased response time might be associated with reduced perceptual detection threshold is to vary the tactile stimulus intensity, and examine whether the addition of auditory stimulus would reduce the detection failure for weak tactile stimuli.”*

The authors missed the key point in their response. The key point is to determine the behavior effect of concurrent auditory stimulation on threshold-level tactile detection. However, the authors only examined the effect of varying stimulus intensity on the activity of L2/3 neurons but not on behavioral performance. The more important question of whether multi-sensory input in the current behavioral task has any effect on the performance of tactile detection remains unaddressed.

We agree that the added experiments, which investigated the influence of different sensory intensities on dendritic integration and were requested by Reviewer 1, do not directly address whether auditory stimulation alters threshold-level tactile detection. Although we agree testing the influence of auditory input on threshold-level tactile detection during behavior would be a useful addition, we have already shown that there is a behavioral impact of pairing auditory input with a tactile stimulus, therefore it is not clear how much value these additional (extremely time consuming) experiments would be, especially considering the work involved. We include the following statement in the revised manuscript: “Pairing an auditory stimulus with a threshold-level tactile stimulus is likely to increase behavioural performance by reducing the perceptual detection threshold, however, the important question of whether multi-sensory input during the behavioral task has any effect on the performance of tactile detection remains unaddressed, and will lead to exciting new research.”

5. *In my previous comments (point 6), when I was asking about the somatic responses, I was referring to the data in Fig. 5, which is from behavioral task. But the new data of somatic activity shown in Supplementary Fig 6 are from passive stimulation. It is important to show that multi-sensory stimulation enhances somatic activity during a task where such multi-sensory information is relevant, as also mentioned above.*

As stated above, we have now performed new experiments where we have recorded the activity of L2/3 somata during behavior (see new Supplementary Figure 11). Here, similar to tuft dendrites, somatic activity is enhanced during AudTac-trials.

Reviewer #3 (Remarks to the Author):

The authors have addressed the various points of critique well and I find the revised manuscript much improved. They have answered most of my questions sufficiently but I still like to make two remarks:

We thank the reviewer for their comments and suggestions and have now included more information in the manuscript to address the concerns.

Ad 3) I find it puzzling that, when varying the loudness of the auditory stimulation, the lowest sound intensity has the largest effect. The authors should at least comment on this peculiar finding and provide their interpretation.

We agree it is puzzling that the lowest sound intensity had the greatest effect on encoding of the tactile stimulus. We now provide an explanation in the discussion, which highlights that the encoding of sound in the auditory cortex is heavily modulated by inhibition. It is possible that the larger sound intensities evoke more local inhibition in the auditory cortex. This remains to be tested, and presents an interesting finding that could be followed up in subsequent studies.

Ad 5) The authors did not answer my question regarding the distribution across FL and HL areas and the comparison with barrel cortex. Perhaps they still have not consulted the Allen atlas in detail, which they should do. They should pull out all the relevant Allen experiments (projections from A1 source to FL/HL/BC targets), look at the 3D histology datasets and convey to the reader what they find in terms of axonal projection patterns across areas and layers. It is also easy to extract from the Allen atlas the quantifications for these projections in terms of axonal fluorescence in the target regions.

We have now included information in the revised manuscript regarding the auditory cortex axonal projection patterns in other regions of the primary somatosensory cortex, including the forelimb (FL), hindlimb (HL) and barrel cortex (BC). In brief, according to the Allen atlas, A1 projects to FL/HL and BC. BC appears to receive projections distributed across all of the cortical layers, whereas FL and HL receive similar projections that are more restricted in their distribution across the layers. Therefore, the influence of auditory stimuli on the sensory processing in the different sensory cortices may differ and we now discuss the different projection pathways and what this might mean for multisensory encoding in the revised manuscript.

REVIEWER COMMENTS

Reviewer #2 (Remarks to the Author):

In this revision, the authors have performed new experiments based on my suggestions, and have addressed most of my concerns. Although not performing the behavioral experiments to test the effect of multi-sensory integration on perceptual detection makes this study less compelling and satisfying, the author has given their reasonable explanation. Therefore, I would like to support the publication of this revised manuscript.

Reviewer #3 (Remarks to the Author):

The authors have adequately responded to my requests.

REVIEWERS' COMMENTS

Reviewer #2 (Remarks to the Author):

In this revision, the authors have performed new experiments based on my suggestions, and have addressed most of my concerns. Although not performing the behavioral experiments to test the effect of multi-sensory integration on perceptual detection makes this study less compelling and satisfying, the author has given their reasonable explanation. Therefore, I would like to support the publication of this revised manuscript.

We thank the reviewer for their helpful comments throughout the resubmission process.

Reviewer #3 (Remarks to the Author):

The authors have adequately responded to my requests.

We thank the reviewer for their helpful comments throughout the resubmission process.